# Singleshot : a scalable Tucker tensor decomposition

**Abraham Traoré**
LITIS EA4108
University of Rouen Normandy
`abraham.traore@etu.univ-rouen.fr`

**Maxime Bérar**
LITIS EA4108
University of Rouen Normandy
`maxime.berar@univ-rouen.fr`

**Alain Rakotomamonjy**
LITIS EA4108
University of Rouen Normandy
Criteo AI Lab, Criteo Paris
`alain.rakoto@insa-rouen.fr`

## Abstract

This paper introduces a new approach for the scalable *Tucker* decomposition problem. Given a tensor $\mathcal{X}$, the algorithm proposed, named *Singleshot*, allows to perform the inference task by processing one subtensor drawn from $\mathcal{X}$ at a time. The key principle of our approach is based on the recursive computations of the gradient and on cyclic update of the latent factors involving only one single step of gradient descent. We further improve the computational efficiency of *Singleshot* by proposing an inexact gradient version named *Singleshotinexact*. The two algorithms are backed with theoretical guarantees of convergence and convergence rates under mild conditions. The scalabilty of the proposed approaches, which can be easily extended to handle some common constraints encountered in tensor decomposition (*e.g* non-negativity), is proven via numerical experiments on both synthetic and real data sets.

## 1   Introduction

The recovery of information-rich and task-relevant variables hidden behind observation data (commonly referred to as latent variables) is a fundamental task that has been extensively studied in machine learning. In many applications, the dataset we are dealing with naturally presents different modes (or dimensions) and thus, can be naturally represented by multidimensional arrays (also called tensors). The recent interest for efficient techniques to deal with such datasets is motivated by the fact that the methodologies that matricize the data and then apply matrix factorization give a flattened view of data and often cause a loss of the internal structure information. Hence, to mitigate the extent of this loss, it is more favorable to process a multimodal data set in its own domain, i.e. tensor domain, to obtain a multiple perspective view of data rather than a flattened one.

Tensors represent generalization of matrices and the related decomposition techniques are promising tools for exploratory analysis of multidimensional data in diverse disciplines including signal processing [11], social networks analysis [28], etc. The two most common decompositions used for tensor analysis are the *Tucker* decomposition [43] and the Canonical Polyadic Decomposition also named *CPD*[16, 6]. These decompositions are used to infer multilinear relationships from multidimensional datasets as they allow to extract hidden (latent) components and investigate the relationships among them.

In this paper, we focus on the *Tucker* decomposition motivated by the fact that this decomposition and its variants have been successfully used in many real applications [24, 19]. Our technical goal

is to develop a scalable *Tucker* decomposition technique in a *static setting* (the tensor is fixed). Such an objective is relevant in a situation where it is not possible to load in memory the tensor of interest or when the decomposition process may result in memory overflow generated by intermediate computations [20, 31].

## 1.1 Related work and main limitations

Divide-and-conquer type methods (i.e. which divide the data set into sub-parts) have already been proposed for the scalable *Tucker* decomposition problem, with the goal of efficiently decomposing a large fixed tensor (*static setting*). There are mainly three trends for these methods: distributed methods, sequential processing of small subsets of entries drawn from the tensor or the computation of the tensor-matrix product in a piecemeal fashion by adaptively selecting the order of the computations. A variant of the *Tucker-ALS* has been proposed in [31] and it solves each alternate step of the Tucker decomposition by processing on-the-fly intermediate data, reducing then the memory footprint of the algorithm. Several other approaches following the same principles are given in [5, 9, 4, 33] while others consider some sampling strategies [29, 36, 14, 39, 48, 18, 47, 35, 27, 10, 25] or distributed approaches [49, 7, 34]. One major limitation related to these algorithms is their lack of genericness (i.e. they cannot be extended to incorporate some constraints such as non-negativity).

Another set of techniques for large-scale *Tucker* decomposition in a static setting focuses on designing both deterministic and randomized algorithms in order to alleviate the computational burden of the decomposition. An approach proposed by [4] performs an alternate minimization and reduces the scale of the intermediate problems via the incorporation of sketching operators. In the same flavor, one can reduce the computational burden of the standard method HOSVD through randomization and by estimating the orthonormal basis via the so-called range finder algorithm [51]. This class of approaches encompasses other methods that can be either random [8, 30, 13, 37, 42, 46] or deterministic [40, 2, 38, 3, 50, 17, 26, 32]. The main limitation of these methods essentially stems from the fact that they use the whole data set at once (instead of dividing it), which makes them non-applicable when the tensor does not fit the available memory.

From a theoretical point of view, among all these works, some algorithms are backed up with convergence results [4] or have quality of approximation guarantees materialized by a recovery bound [1]. However, there is still a lack of convergence rate analysis for the scalable *Tucker* problem.

## 1.2 Main contributions

In contrast to the works described above, our contributions are the following ones:

- We propose a new approach for the scalable *Tucker* decomposition problem, denoted as *Singleshot* leveraging on coordinate gradient descent [41] and sequential processing of data chunks amenable to constrained optimization.
- In order to improve the computational efficiency of *Singleshot*, we introduce an inexact gradient variant, denoted as *Singleshotinexact*. This inexact approach can be further extended so as to make it able to decompose a tensor growing in every mode and in an online fashion.
- From a theoretical standpoint, we establish for *Singleshot* an ergodic convergence rate of $\mathcal{O}\left(\frac{1}{\sqrt{K}}\right)$ ($K$: maximum number of iterations) to a stationary point and for *Singleshotinexact*, we establish a convergence rate of $\mathcal{O}(\frac{1}{k})$ ($k$ being the iteration number) to a minimizer.
- We provide experimental analyses showing that our approaches are able to decompose bigger tensors than competitors without compromising efficiency. From a streaming tensor decomposition point of view, our *Singleshot* extension is competitive with its competitor.

## 2 Notations & Definitions

A N−order tensor is denoted by a boldface Euler script letter $\boldsymbol{\mathcal{X}} \in \mathbb{R}^{I_1 \times \cdots \times I_N}$. The matrices are denoted by bold capital letters (e.g. $\mathbf{A}$). The identity matrix is denoted by $\mathbf{Id}$. The $j^{th}$ row of a matrix $\mathbf{A} \in \mathbb{R}^{J \times L}$ is denoted by $\mathbf{A}_{j,:}$ and the transpose of a matrix $\mathbf{A}$ by $\mathbf{A}^{\top}$.
Matricization is the process of reordering all the elements of a tensor into a matrix. The mode-$n$ matricization of a tensor $[\boldsymbol{\mathcal{X}}]^{(n)}$ arranges the mode-$n$ fibers to be the columns of the resulting matrix $\mathbf{X}^{(n)} \in \mathbb{R}^{I_n \times (\prod_{m \neq n} I_m)}$. The mode-$n$ product of a tensor $\boldsymbol{\mathcal{G}} \in \mathbb{R}^{J_1 \times \cdots \times J_N}$ with a matrix

$\mathbf{A} \in \mathbb{R}^{I_n \times J_n}$ denoted by $\mathcal{G} \times_n \mathbf{A}$ yields a tensor of the same order $\mathcal{B} \in \mathbb{R}^{J_1 \times \cdots J_{n-1} \times I_n \times J_{n+1} \cdots \times J_N}$ whose mode-$n$ matricized form is defined by: $\mathbf{B}^{(n)} = \mathbf{A}\mathbf{G}^{(n)}$. For a tensor $\mathcal{X} \in \mathbb{R}^{I_1 \times \cdots \times I_N}$, its $i_n^{th}$ subtensor with respect to the mode $n$ is denoted by $\mathcal{X}_{i_n}^n \in \mathbb{R}^{I_1 \times \cdots \times I_{n-1} \times 1 \times I_{n+1} \times \cdots \times I_N}$. This subtensor is a $N$-order tensor defined via the mapping between its $n$-mode matricization $\left[\mathcal{X}_{i_n}^n\right]^{(n)}$ and the $i_n^{th}$ row of $\mathbf{X}^{(n)}$, i.e. the tensor $\mathcal{X}_{i_n}^n$ is obtained by reshaping the $i_n^{th}$ row of $\mathbf{X}^{(n)}$, with the target shape $(I_1, .., I_{n-1}, 1, I_{n+1}, .., I_N)$. The set of integers from $n$ to $N$ is denoted by $\mathbf{I}_N^n = \{n, .., N\}$: if $n = 1$, the set is simply denoted by $\mathbf{I}_N$. The set of integers from 1 to $N$ with $n$ excluded is denoted by $\mathbf{I}_{N \neq n} = \{1, .., n-1, n+1, .., N\}$. Let us define the tensor $\mathcal{G} \in \mathbb{R}^{J_1 \times \cdots \times J_N}$ and $\mathbf{N}$ matrices $\left\{\mathbf{A}^{(m)} \in \mathbb{R}^{I_m \times J_m}\right\}_{n \in \mathbf{I}_N}$. The product of $\mathcal{G}$ with the matrices $\mathbf{A}^{(m)}, m \in \mathbf{I}_N$ denoted by $\mathcal{G} \times_1 \mathbf{A}^{(1)} \times_2 ... \times_N \mathbf{A}^{(N)}$ will be alternatively expressed by:

$$\mathcal{G} \underset{m \in \mathbf{I}_N}{\times_m} \mathbf{A}^{(m)} = \mathcal{G} \underset{m \in \mathbf{I}_{n-1}}{\times_m} \mathbf{A}^{(m)} \times_n \mathbf{A}^{(n)} \underset{q \in \mathbf{I}_N^{n+1}}{\times_q} \mathbf{A}^{(q)} = \mathcal{G} \underset{m \in \mathbf{I}_{N \neq n}}{\times_m} \mathbf{A}^{(m)} \times_n \mathbf{A}^{(n)}.$$

The Frobenius norm of a tensor $\mathcal{X} \in \mathbb{R}^{I_1 \times \cdots \times I_N}$, denoted by $\|\mathcal{X}\|_F$ is defined by: $\|\mathcal{X}\|_F = \left(\sum_{1 \leq i_n \leq I_n, 1 \leq n \leq N} \mathcal{X}_{i_1, \cdots, i_N}^2\right)^{\frac{1}{2}}$. The same definition holds for matrices.

# 3 Piecewise tensor decomposition: *Singleshot*

## 3.1 *Tucker* decomposition and problem statement

Given a tensor $\mathcal{X} \in \mathbb{R}^{I_1 \times \cdots \times I_N}$, the *Tucker* decomposition aims at the following approximation:

$$\mathcal{X} \approx \mathcal{G} \underset{m \in \mathbf{I}_N}{\times_m} \mathbf{A}^{(m)}, \mathcal{G} \in \mathbb{R}^{J_1 \times \cdots \times J_N}, \mathbf{A}^{(m)} \in \mathbb{R}^{I_m \times J_m}$$

The tensor $\mathcal{G}$ is generally named the core tensor and the matrices $\left\{\mathbf{A}^{(m)}\right\}_{m \in \mathbf{I}_N}$ the loading matrices. With orthogonality constraints on the loading matrices, this decomposition can be seen as the multidimensional version of the singular value decomposition [23].
A natural way to tackle this problem is to infer the latent factors $\mathcal{G}$ and $\left\{\mathbf{A}^{(m)}\right\}_{m \in \mathbf{I}_N}$ in such a way that the discrepancy is low. Thus, the decomposition of $\mathcal{X}$ is usually obtained by solving the following optimization problem:

$$\min_{\mathcal{G}, \mathbf{A}^{(1)}, \cdots, \mathbf{A}^{(N)}} \left\{ f\left(\mathcal{G}, \mathbf{A}^{(1)}, \cdots, \mathbf{A}^{(N)}\right) \triangleq \frac{1}{2}\|\mathcal{X} - \mathcal{G} \times_{m \in \mathbf{I}_N} \mathbf{A}^{(m)}\|_F^2 \right\} \tag{1}$$

Our goal in this work is to solve the above problem, for large tensors, while addressing two potential issues : the processing of a tensor that does not fit into the available memory and avoiding memory overflow problem generated by intermediate operations during the decomposition process [21].

For this objective, we leverage on a reformulation of the problem (1) in terms of subtensors drawn from $\mathcal{X}$ with respect to one mode (which we suppose to be predefined), the final objective being to set up a divide-and-conquer type approach for the inference task. Let's consider a fixed integer $n$ (in the sequel, $n$ will be referred to as the splitting mode). Indeed, the objective function can be rewritten in the following form (see supplementary, property 2):

$$f\left(\mathcal{G}, \mathbf{A}^{(1)}, \cdots, \mathbf{A}^{(N)}\right) = \sum_{i_n=1}^{I_n} \frac{1}{2}\|\mathcal{X}_{i_n}^n - \mathcal{G} \underset{m \in \mathbf{I}_{N \neq n}}{\times_m} \mathbf{A}^{(m)} \times_n \mathbf{A}_{i_n,:}^{(n)}\|_F^2 \tag{2}$$

More generally, the function $f$ given by (1) can be expressed in terms of subtensors drawn with respect to every mode (see supplementary material, property 3). For simplicity concerns, we only address the case of subtensors drawn with respect to one mode and the general case can be derived following the same principle (see supplementary material, section 5).

## 3.2 Singleshot

Since the problem (1) does not admit any analytic solution, we propose a numerical resolution based on coordinate gradient descent [41]. The underlying idea is based on a cyclic update over each of the variables $\mathcal{G}, \mathbf{A}^{(1)}, .., \mathbf{A}^{(N)}$ while fixing the others at their last updated values and each

---
**Algorithm 1** *Singleshot*

---

**Inputs**:        $\mathcal{X}$ tensor of interest, $n$ splitting mode, $\left\{\mathbf{A}_0^{(m)}\right\}_{1\leq m\leq N}$ initial loading matrices,

**Output**:        $\mathcal{G},\left\{\mathbf{A}^{(m)}\right\}_{1\leq m\leq N}$

**Initialization**:   $k=0$

1: **while** a predefined stopping criterion is not met **do**
2:     Compute optimal step $\eta_k^{\mathcal{G}}$
3:     $\mathcal{G}_{k+1} \leftarrow \mathcal{G}_k - \eta_k^{\mathcal{G}}\mathcal{D}_k^{\mathcal{G}}$                             with $\mathcal{D}_k^{\mathcal{G}}$ given by (4)
4:     **for** p from 1 to N **do**
5:        Compute optimal step $\eta_k^p$
6:        $\mathbf{A}_{k+1}^{(p)} \leftarrow \mathbf{A}_k^{(p)} - \eta_k^p \mathbf{D}_k^p$                     with $\mathbf{D}_k^p$ given by (5),(6)
7:     **end for**
8: **end while**

---

update being performed via a single iteration of gradient descent. More formally, given at iteration $k$, $\mathcal{G}_k, \mathbf{A}_k^{(1)}, ..., \mathbf{A}_k^{(N)}$ the value of the latent factors, the derivatives $\mathcal{D}_k^{\mathcal{G}}$ and $\mathbf{D}_k^p$ of $f$ with respect to the core tensor and the $p^{th}$ loading matrix respectively evaluated at $\left(\mathcal{G}_k, \mathbf{A}_k^{(1)}, \cdots, \mathbf{A}_k^{(N)}\right)$ and $\left(\mathcal{G}_{k+1}, \mathbf{A}_{k+1}^{(1)}, \cdots, \mathbf{A}_{k+1}^{(p-1)}, \mathbf{A}_k^{(p)}., \mathbf{A}_k^{(N)}\right)$ are given by:

$$\mathcal{D}_k^{\mathcal{G}} = \partial_{\mathcal{G}} f\left(\mathcal{G}_k, \left\{\mathbf{A}_k^{(m)}\right\}_1^N\right), \qquad \mathbf{D}_k^p = \partial_{\mathbf{A}^{(p)}} f\left(\mathcal{G}_{k+1}, \left\{\mathbf{A}_{k+1}^{(m)},\right\}_1^{p-1}, \left\{\mathbf{A}_k^{(q)}\right\}_p^N\right) \quad (3)$$

The resulting cyclic update algorithm, named *Singleshot*, is summarized in Algorithm 1. A naive implementation of the gradient computation would result in memory overflow problem. In what follows, we show that the derivatives $\mathcal{D}_k^{\mathcal{G}}$ and $\mathbf{D}_k^p, 1 \leq p \leq N$ given by the equation (3) can be computed by processing a single subtensor at a time, making Algorithm 1 amenable to sequential processing of subtensors. Discussions on how the step sizes are obtained will be provide in Section 4.

**Derivative with respect to $\mathcal{G}$.** The derivative with respect to the core tensor is given by (details in Property 7 of supplementary material):

$$\mathcal{D}_k^{\mathcal{G}} = \sum_{i_n=1}^{I_n} \underbrace{\mathcal{R}_{i_n} \underset{m\in\mathbf{I}_{N\neq n}}{\times_m} \left(\mathbf{A}_k^{(m)}\right)^\top \times_n \left(\left(\mathbf{A}_k^{(n)}\right)_{i_n,:}\right)^\top}_{\theta_{i_n}}, \mathcal{R}_{i_n} = -\mathcal{X}_{i_n}^n + \mathcal{G}_k \underset{m\in\mathbf{I}_{N\neq n}}{\times_m} \mathbf{A}_k^{(m)} \times_n \left(\mathbf{A}_k^{(n)}\right)_{i_n,:}$$

$$(4)$$

It is straightforward to see that $\mathcal{D}_k^{\mathcal{G}}$ (given by the equation (4)) is the last term of the recursive sequence $\left\{(\mathcal{D}_k^{\mathcal{G}})^j\right\}_{1\leq j\leq I_n}$ defined as $\left(\mathcal{D}_k^{\mathcal{G}}\right)^j = \left(\mathcal{D}_k^{\mathcal{G}}\right)^{j-1} + \theta_j$, with $\left(\mathcal{D}_k^{\mathcal{G}}\right)^0$ being the null tensor. An important observation is that the additive term $\theta_j$ (given by the equation (4)) depends only on one single subtensor $\mathcal{X}_j^n$. This is the key of our approach since it allows the computation of $\mathcal{D}_k^{\mathcal{G}}$ through the sequential processing of a single subtensor $\mathcal{X}_j^n$ at a time.

**Derivatives with respect to $\mathbf{A}^{(p)}, p \neq n$ ($n$ being the splitting mode).** For those derivatives, we can exactly follow the same reasoning, given in detail in Property 9 of the Supplementary material, and obtain for $p < n$ (the case $p > n$ yields a similar formula):

$$\mathbf{D}_k^p = \sum_{i_n=1}^{I_n} \left(-\left(\mathbf{X}_{i_n}^n\right)^{(p)} + \mathbf{A}_k^{(p)}\mathbf{B}_{i_n}^{(p)}\right)\left(\mathbf{B}_{i_n}^{(p)}\right)^\top \qquad (5)$$

The matrices $(\mathbf{X}_{i_n}^n)^{(p)}$ and $\mathbf{B}_{i_n}^{(p)}$ represent respectively the mode-$p$ matricized forms of the $i_n^{th}$ subtensor $\mathcal{X}_{i_n}^n$ and the tensor $\mathcal{B}_{i_n}$ is defined by:

$$\mathcal{B}_{i_n} = \mathcal{G}_{k+1} \underset{m\in\mathbf{I}_{p-1}}{\times_m} \mathbf{A}_{k+1}^{(m)} \times_p \mathbf{Id} \underset{q\in\mathbf{I}_{N\neq n}^{p+1}}{\times_q} \mathbf{A}_k^{(q)} \times_n \left(\mathbf{A}_k^{(n)}\right)_{i_n,:}$$

with $\mathbf{Id} \in \mathbb{R}^{J_p \times J_p}$ being the identity matrix. With a similar reasoning as for the derivative with respect to the core, it is straightforward to see that $\mathbf{D}_k^p$ can be computed by processing a single subtensor at a time.

**Derivative with respect to $\mathbf{A}^{(n)}$ ($n$ being the splitting mode).** The derivative with respect to the matrix $\mathbf{A}^{(n)}$ can be computed via the row-wise stacking of independent terms, that are, the derivatives with respect to the rows $\mathbf{A}_{j,:}^{(n)}$ and the derivative of $f$ with respect to $\mathbf{A}_{j,:}^{(n)}$ depends only on $\mathcal{X}_j^n$. Indeed, let's consider $1 \leq j \leq I_n$. In the expression of the objective function $f$ given by the equation (2), the only term that depends on $\mathbf{A}_{j,:}^{(n)}$ is $\|\mathcal{X}_j^n - \mathcal{G} \underset{m \in \mathbf{I}_{n-1}}{\times_m} \mathbf{A}^{(m)} \times_n \mathbf{A}_{j,:}^{(n)} \underset{q \in \mathbf{I}_N^{n+1}}{\times_q} \mathbf{A}^{(q)}\|_F^2$, thus

the derivative of $f$ with respect to $\mathbf{A}_{j,:}^{(n)}$ depends only on $\mathcal{X}_j^n$ and is given by (see property 8 in the supplementary material):

$$\partial_{\mathbf{A}_{j,:}^{(n)}} f\left(\mathcal{G}, \left\{\mathbf{A}^{(m)}\right\}_1^N\right) = -\left((\mathbf{X}_j^n)^{(n)} - \mathbf{A}_{j,:}^{(n)} \mathbf{B}^{(n)}\right) \mathbf{B}^{(n)\top} \tag{6}$$

The tensors $(\mathbf{X}_j^n)^{(n)} \in \mathbb{R}^{1 \times \prod_{k \neq n} I_k}$ and $\mathbf{B}^{(n)}$ respectively represent the mode-n matricized form of the tensors $\mathcal{X}_j^n$ and $\mathcal{B}$ with $\mathcal{B} = \mathcal{G} \underset{p \in \mathbf{I}_{n-1}}{\times_p} \mathbf{A}^{(p)} \times_n \mathbf{Id} \underset{q \in \mathbf{I}_N^{n+1}}{\times_q} \mathbf{A}^{(q)}, \mathbf{Id} \in \mathbb{R}^{J_n \times J_n}$: identity matrix.

**Remark 1.** *. For one-mode subtensors, it is relevant to choose $n$ such that $I_n$ is the largest dimension since this yields the smallest subtensors. We stress that all entries of the tensor $\mathcal{X}$ have been entirely processed when running Algorithm 1 and our key action is the sequential processing of subtensors $\mathcal{X}_{i_n}^n$. In addition, if one subtensor does not fit in the available memory, the recursion, as shown in section 5 of the supplementary material, can still be applied to subtensors of the form $\mathcal{X}_{\theta_1,...,\theta_N}, \theta_m \subset \{1, 2, .., I_m\}$ with $(\mathcal{X}_{\theta_1,..,\theta_N})_{i_1,..,i_N} = \mathcal{X}_{i_1,..,i_N}, (i_1,..,i_N) \in \theta_1 \times ... \times \theta_N, \times$ referring to the Cartesian product.*

### 3.3 *Singleshotinexact*

While all of the subtensors $\mathcal{X}_{i_n}^n, 1 \leq i_n \leq I_n$ are considered in the *Singleshot* algorithm for the computation of the gradient, in *Singleshotinexact*, we propose to use only a subset of them for the sake of reducing computational time. The principle is to use for the gradients computation only $B_k < I_n$ subtensors. Let's consider the set $\mathcal{SET}_k$ (of cardinality $B_k$) composed of the integers representing the indexes of the subtensors that are going to be used at iteration $k$. The numerical resolution scheme is identical to the one described by Algorithm 1 except for the definition of $\mathcal{D}_k^{\mathcal{G}}$ and $\mathbf{D}_k^p$ which are respectively replaced by $\widehat{\mathcal{D}}_k^{\mathcal{G}}$ and $\widehat{\mathbf{D}}_k^p, p \neq n$ defined by:

$$\widehat{\mathcal{D}}_k^{\mathcal{G}} = \sum_{i_n \in \mathcal{SET}_k} \mathcal{R}_{i_n} \underset{m \in \mathbf{I}_{N \neq n}}{\times_m} \mathbf{A}_k^{(m)\top} \times_n \left(\left(\mathbf{A}_k^{(n)}\right)_{i_n,:}\right)^\top \tag{7}$$

$$\widehat{\mathbf{D}}_k^p = \sum_{i_n \in \mathcal{SET}_k} \left(-\left(\mathbf{X}_{i_n}^n\right)^{(p)} + \mathbf{A}^{(p)} \mathbf{B}_{i_n}^{(p)}\right) \left(\mathbf{B}_{i_n}^{(p)}\right)^\top \tag{8}$$

For the theoretical convergence, the descent steps are defined as $\frac{\eta_k^{\mathcal{G}}}{B_k}$ and $\frac{\eta_k^p}{B_k}, 1 \leq p \leq N$. It is worth to highlight that the derivative $\widehat{\mathbf{D}}_k^n$ ($n$ being the mode with respect to which the subtensors are drawn) is sparse: *Singleshotinexact* amounts to minimize $f$ defined by (2) by dropping the terms $\left\{\|\mathcal{X}_j^n - \mathcal{G} \underset{m \in \mathbf{I}_{N \neq n}}{\times_m} \mathbf{A}^{(m)} \times_n \mathbf{A}_{j,:}^{(n)}\|_F^2\right\}$ with $j \notin \mathcal{SET}_k$, thus, the rows $\left(\widehat{\mathbf{D}}_k^n\right)_{j,:}, j \notin \mathcal{SET}_k$ are all equal to zero.

### 3.4 Discussions

First, we discuss the space complexity needed by our algorithms supposing that the subtensors are drawn with respect to one mode. Let's denote by $n$ the splitting mode. For *Singleshot* and *Singleshot-inexact*, at the same time, we only need to have in memory the tensor $\mathcal{X}_j^n$ of size

$\prod_{m \in \mathbf{I}_{N \neq n}} I_m = I_1..I_{n-1}I_{n+1}..I_N$, the matrices $\left\{\mathbf{A}^{(m)}\right\}_{m \in \mathbf{I}_{N \neq n}}$, $\mathbf{A}^{(n)}_{i_n,:}$ and the previous iterate of the gradient. Thus, the complexity in space is $\prod_{m \in \mathbf{I}_{N \neq n}} I_m + \sum_{m \neq n} I_m J_m + J_n + \mathcal{AT}$ with $\mathcal{AT}$ being the space complexity of the previous gradient iterate: for the core update, $\mathcal{AT} = \prod_{m \in \mathbf{I}_N} J_m$ and for a matrix $\mathbf{A}^{(m)}$, $\mathcal{AT} = I_m J_m$. If the recursion used for the derivatives computation is applied to subtensors of the form $\mathcal{X}_{\theta_1, \cdots, \theta_N}$, the space complexity is smaller than these complexities.

Another variant of *Singleshotinexact* can be derived to address an interesting problem that has received little attention so far [4], that is the decomposition of a tensor streaming in every mode with a single pass constraint (i.e. each chunk of data is processed only once) named *Singleshotonline*. This is enabled by the inexact gradient computation which uses only subtensors that are needed. In the streaming context, the gradient is computed based only on the available subtensor.

Positivity constraints is one of the most encountered constraints in tensor computation and we can simply handle those constraints via the so-called projected gradient descent [45]. This operation does not alter the space complexity with respect to the unconstrained case, since no addition storage is required but increases the complexity in time. For more details, see the section 3 in the supplementary material for the algorithmic details for the proposed variants.

## 4 Theoretical result

Let's consider the minimization problem (1):

$$\min_{\mathcal{G}, \mathbf{A}^{(1)}, ..., \mathbf{A}^{(N)}} f\left(\mathcal{G}, \mathbf{A}^{(1)}, .., \mathbf{A}^{(N)}\right)$$

By denoting the block-wise derivative by $\partial_{\mathbf{x}} f$, the derivative of $f$, denoted $\nabla f$ and defined by $(\partial_{\mathcal{G}} f, \partial_{\mathbf{A}^{(1)}} f..\partial_{\mathbf{A}^{(N)}} f)$, is an element of $\mathbb{R}^{J_1 \times .. \times J_N} \times \mathbb{R}^{I_1 \times J_1} \times ... \times \mathbb{R}^{I_N \times J_N}$ endowed with the norm $\|\cdot\|_*$ defined as the sum of the Frobenius norms. Besides, let's consider, for writing simplicity, the alternative notations of $f(\mathcal{G}, \mathbf{A}^{(1)}, \cdots, \mathbf{A}^{(N)})$ given by: $f(\mathcal{G}, \left\{\mathbf{A}^{(m)}\right\}_1^N)$, $f(\mathcal{G}, \left\{\mathbf{A}^{(m)}\right\}_1^p, \left\{\mathbf{A}^{(q)}\right\}_{p+1}^N)$. For the theoretical guarantees which details have been reported in the supplementary material, we consider the following assumptions:

**Assumption 1.** *Uniform boundedness. The $n^{th}$ subtensors are bounded:* $\|\mathcal{X}^n_{i_n}\|_F \leq \rho$.

**Assumption 2.** *Boundedness of factors. We consider the domain $\mathcal{G} \in \mathbb{D}_g$, $\mathbf{A}^{(m)} \in \mathbb{D}_m$ with:*

$$\mathbb{D}_g = \{\|\mathcal{G}_a\|_F \leq \alpha\}, \mathbb{D}_m = \left\{\|\mathbf{A}^{(m)}_a\|_F \leq \alpha\right\}$$

### 4.1 Convergence result of *Singleshot*

For the convergence analysis, we consider the following definitions of the descent steps $\eta_k^{\mathcal{G}}$ and $\eta_k^p$ at the $(k+1)^{th}$ iteration:

$$\eta_k^{\mathcal{G}} = \operatorname*{arg\,min}_{\eta \in [\frac{\delta_1}{\sqrt{K}}, \frac{\delta_2}{\sqrt{K}}]} (\eta - \frac{\delta_1}{\sqrt{K}})\phi_g(\eta), \eta_k^p = \operatorname*{arg\,min}_{\eta \in [\frac{\delta_1}{\sqrt{K}}, \frac{\delta_2}{\sqrt{K}}]} (\eta - \frac{\delta_1}{\sqrt{K}})\phi_p(\eta) \qquad (9)$$

$$\phi_g(\eta) = f\left(\mathcal{G}_k - \eta \mathcal{D}_k^{\mathcal{G}}, \left\{\mathbf{A}_k^{(m)}\right\}_1^N\right) - f\left(\mathcal{G}_k, \left\{\mathbf{A}_k^{(m)}\right\}_1^N\right)$$

$$\phi_p(\eta) = f\left(\mathcal{G}_{k+1}, \left\{\mathbf{A}_{k+1}^{(m)}\right\}_1^{p-1}, \mathbf{A}_k^{(p)} - \eta \mathbf{D}_k^p, \left\{\mathbf{A}_k^{(q)}\right\}_{p+1}^N\right) - f\left(\mathcal{G}_{k+1}, \left\{\mathbf{A}_{k+1}^{(m)}\right\}_1^{p-1}, \left\{\mathbf{A}_k^{(q)}\right\}_p^N\right)$$

and $\delta_2 > \delta_1 > 0$ being user-defined parameters. The motivation of the problems given by the equation (9) is to ensure a decreasing of the objective function after each update. Also note that, the minimization problems related to $\eta_k^{\mathcal{G}}$ and $\eta_k^p$ are well defined since all the factors involved in their definitions are known at the computation stage of $\mathcal{G}_{k+1}$ and $\mathbf{A}_{k+1}^{(p)}$ and correspond to the minimization of a continuous function on a compact set.

Along with Assumption 1 and Assumption 2, as well as the definitions given by (9), we assume that:

$$\frac{\delta_1}{\sqrt{K}} < \eta_k^{\mathcal{G}} \leq \frac{\delta_2}{\sqrt{K}} \quad \text{and} \quad \frac{\delta_1}{\sqrt{K}} < \eta_k^p \leq \frac{\delta_2}{\sqrt{K}} \tag{10}$$

This simply amounts to consider that the solutions of the minimization problems defined by the equation (9) are not attained at the lower bound of the interval. Under this framework, we establish, as in standard non-convex settings [12], an ergodic convergence rate. Precisely, we prove that:

$$\exists K_0 \geq 1, \forall K \geq K_0, \frac{1}{K} \sum_{k=0}^{K-1} \|\nabla f\left(\mathcal{G}_k, \left\{\mathbf{A}_k^{(m)}\right\}_1^N\right)\|_*^2 \leq \frac{(N+1)\Delta}{\sqrt{K}} \tag{11}$$

with $\|\nabla f\left(\mathcal{G}_k, \left\{\mathbf{A}_k^{(m)}\right\}_1^N\right)\|_* = \|\partial_{\mathcal{G}} f(\mathcal{G}_k, \left\{\mathbf{A}_k^{(m)}\right\}_1^N)\|_F + \sum_{p=1}^N \|\partial_{\mathbf{A}^{(p)}} f(\mathcal{G}_k, \left\{\mathbf{A}_k^{(m)}\right\}_1^N)\|_F$,
$\Delta = \frac{\delta_2}{\delta_1^2}\left(2\Gamma + \alpha^{2N}\Gamma_g^2\delta_2^2 + \sum_{p=1}^N(1 + 2\Gamma + \alpha^{2N}\Gamma_p^2\delta_2^2)\right)$, $\Gamma, \Gamma_p, \Gamma_g \geq 0$ being respectively the supremun of $f, \|\partial_{\mathbf{A}^{(p)}} f(\mathcal{G}, \left\{\mathbf{A}^{(m)}\right\})\|_F, \|\partial_{\mathcal{G}} f(\mathcal{G}, \left\{\mathbf{A}^{(m)}\right\})\|_F$ on the compact set $\mathbb{D}_g \times \mathbb{D}_1.. \times \mathbb{D}_N$
This result proves that *Singleshot* converges ergodically to a point where the gradient is equal to 0 at the rate $\mathcal{O}\left(\frac{1}{\sqrt{K}}\right)$.

## 4.2 Convergence result for *Singleshotinexact*

Let us consider that $\ell_j(\mathbf{A}^{(N)}) \triangleq \frac{1}{2}\|\mathcal{X}_j^n - \mathcal{G}_{k+1} \underset{m \in \mathbf{I}_{n-1}}{\times_m} \mathbf{A}_{k+1}^{(m)} \times_n (\mathbf{A}_{k+1}^{(n)})_{j,:} \underset{q \in \mathbf{I}_{N-1}^{n+1}}{\times_q} \mathbf{A}_{k+1}^{(q)} \times_N \mathbf{A}^{(N)}\|_F^2$

and that the step $\eta_k^N$ for $\mathbf{A}^{(N)}$ is defined by the following minimization problem:

$$\eta_k^N = \underset{\eta \in [\frac{1}{4K\gamma}, \frac{1}{K\gamma}]}{\arg\min} \left(\eta - \frac{1}{4K\gamma}\right)\phi(\eta) \tag{12}$$

$$\phi(\eta) = f\left(\mathcal{G}_{k+1}, \left\{\mathbf{A}_{k+1}^{(m)}\right\}_1^{N-1}, \mathbf{A}_k^{(N)}\right) - f\left(\mathcal{G}_k, \left\{\mathbf{A}_k^{(m)}\right\}_1^N\right) + \lambda f\left(\mathcal{G}_{k+1}, \left\{\mathbf{A}_{k+1}^{(m)}\right\}_1^{N-1}, \tilde{\phi}(\eta)\right)$$

and $\tilde{\phi}(\eta) = \mathbf{A}_k^{(N)} - \frac{\eta}{B_k} \sum_{j \in \mathcal{SET}_k} \partial_{\mathbf{A}^{(N)}} \ell_j$, $\partial_{\mathbf{A}^{(N)}} \ell_j$ being the derivative of $\ell_j$ evaluated at $\mathbf{A}_k^{(N)}$.

The parameters $\lambda > 0, \gamma > 1$ represent user-defined parameters. In addition to Assumption 1 and Assumption 2, we consider the following three additional assumptions:

**1. Descent step related to the update of $\mathbf{A}^{(N)}$.** We assume that $\frac{1}{4K\gamma} < \eta_k^N \leq \frac{1}{\alpha^{2N}}$, which means that the solution of problem (12) is not attained at the lower bound of the interval.

**2. Non-vanishing gradient with respect to $\mathbf{A}^{(N)}$** $\partial_{\mathbf{A}^{(N)}} f\left(\mathcal{G}_{k+1}, \left\{\mathbf{A}_{k+1}^{(m)}\right\}_1^{N-1}, \mathbf{A}_k^{(N)}\right) \neq 0$.

This condition ensures the existence of a set $\mathcal{SET}_k$ such that $\sum_{j \in \mathcal{SET}_k} \partial_{\mathbf{A}^{(N)}} \ell_j \neq 0$ and the set considered for the problem (12) is one of such sets.

**3. Choice of the number of subtensors $B_k$.** We suppose that $I_n \times \sqrt{\frac{1}{2} + \frac{1}{I_n}} \leq B_k$ and $I_n > 2$. This condition $I_n > 2$ ensures that $I_n \sqrt{\frac{1}{2} + \frac{1}{I_n}} < I_n$.

With these assumptions at hand, the sequence $\Delta_k = f\left(\mathcal{G}_k, \left\{\mathbf{A}_k^{(m)}\right\}_1^N\right) - f_{\min}$ verifies:

$$\forall k > k_0 = 1 + \frac{1}{\log(1+\lambda)}\log\left(\frac{1}{\log(1+\lambda)}\right), \Delta_k \leq \frac{\Delta_1 + \zeta(\lambda, \rho, \alpha, I_n)}{k - k_0} \tag{13}$$

with log being the logarithmic function, $f_{\min}$ representing the minimizer of $f$, a continuous function defined on the compact set $\mathbb{D}_g \times \mathbb{D}_1 \times .... \times \mathbb{D}_N$ and $\zeta$ a function of $\lambda, \rho, \alpha, I_n$. The parameter $k_0$ is well-defined since $\lambda > 0$. This result ensures that the sequence $\left\{\mathcal{G}_k, \mathbf{A}_k^{(1)}, .., \mathbf{A}_k^{(N)}\right\}$ generated by *Singleshotinexact* converges to the set of minimizers of $f$ at the rate $\mathcal{O}\left(\frac{1}{k}\right)$

**Remark 2.** *The problems defined by the equations* (9) *and* (12)*, which solutions are global and can be solved by simple heuristics (e.g. **Golden section**), are not in contradiction with our approach since they can be solved by processing a single subtensor at a time due to the expression of $f$ given by (2).*

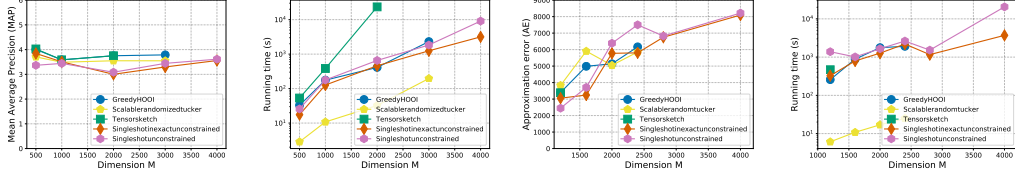

Figure 1: Approximation error and running time for the unconstrained decomposition algorithms. From left to right: first and second figures represent Movielens, third and fourth represent Enron. As $M$ grows , missing markers for a given algorithm means that it ran out of memory. The core $\mathcal{G}$ rank is $(5, 5, 5)$.

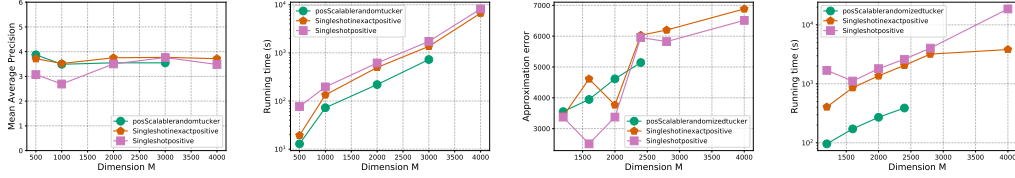

Figure 2: Approximation error and running time for the non-negative decomposition algorithms. From left to right: first and second figures represent Movielens, third and fourth represent Enron. As $M$ grows, missing markers, for a given algorithm means that it ran out of memory. The core $\mathcal{G}$ rank is $(5, 5, 5)$.

# 5 Numerical experiments

Our goal here is to illustrate that for small tensors, our algorithms *Singleshot* and *Singleshotinexact* and their positive variants, are comparable to some state-the-art decomposition methods. Then as the tensor size grows, we show that they are the only ones that are scalable. The competitors we have considered include SVD-based iterative algorithm [44](denoted *GreedyHOOI* ), a very recent alternate minimization approach based on sketching [4] (named *Tensorsketch*) and randomization-based methods [51] (Algorithm 2 in [51] named *Scalrandomtucker* and Algorithm 1 in [51] with positivity constraints named *posScalrandomtucker*). Other materials related to the numerical experiments are provided in the section 4 of the supplementary material. For the tensor computation, we have used the TensorLy tool [22].

The experiments are performed on the *Movielens* dataset [15], from which we construct a 3-order tensor whose modes represent timesteps, movies and users and on the *Enron* email dataset, from which a 3-order tensor is constructed, the first and second modes representing the sender and recipients of the emails and the third one denoting the most frequent words used in the miscellaneous emails. For Movielens, we set up a recommender system for which we report a mean average precision (MAP) obtained over a test set (averaged over five $50 - 50$ train-test random splits) and for Enron, we report an error (AE) on a test set (with the same size as the training one) for a regression problem. As our goal is to analyze the scalability of the different methods as the tensor to decompose grows, we have arbitrarily set the size of the Movielens and Enron tensors to $M \times M \times 200$ and $M \times M \times 610$, $M$ being a user-defined parameter. Experiments have been run on MacOs with 32 Gb of memory.

Another important objective is to prove the robustness of our approach with respect to the assumptions and the definitions related to the descent steps laid out in the section 4, which is of primary importance since the minimization problems defining these steps can be time-consuming in practice for large tensors. This motivates the fact that for our algorithms, the descent steps are fixed in advance. For *Singleshot*, the steps are fixed to $10^{-6}$. For *Singleshot-inexact*, the steps are respectively fixed to $10^{-6}$ and $10^{-8}$ for *Enron* and *Movielens*. Regarding the computing of the inexact gradient for *Singleshotinexact*, the elements in $\mathcal{SET}_k$ are generated randomly without replacement with the same cardinality for any $k$. The number of slices is chosen according to the third assumption in section 4.2. For the charts, the unconstrained versions of *Singleshot* and *Singleshotinexact* will be followed by the term "unconstrained" and "positive" for the positive variantes.

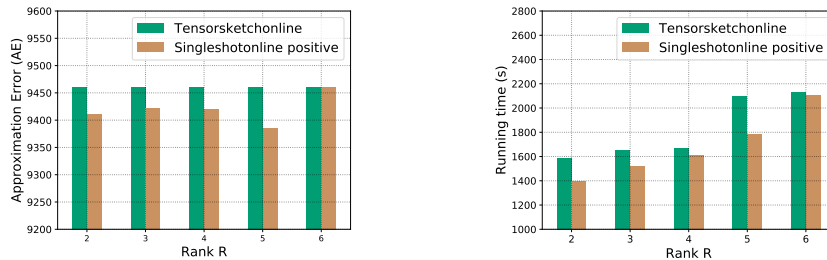

Figure 3: Comparing Online version of Tensorsketch and Singleshot with positive constraints on the Enron dataset. (left) Approximation error. (right) running time.

Figure 1 presents the results we obtain for these two datasets. At first, we can note that performance, in terms of MAP or AE, are rather equivalent for the different methods. Regarding the running time, the *Scalrandomtucker* is the best performing algorithm being an order of magnitude more efficient than other approaches. However, all competing methods struggle to decompose tensors with dimension $M = 4000$ and $M \geq 2800$ respectively for *Enron* and *Movielens* due to memory error. Instead, our Singleshot methods are still able to decompose those tensors, although the running time is large. As expected, *Singleshotinexact* is more computationally efficient than *Singleshot*.

Figure 2 displays the approximation error and the running time for *Singleshot* and *singleshotinexact* with positivity constraints and a randomized decomposition approach with non-negativity constraints denoted here as *PosScalrandomtucker* for *Enron* and *Movielens*. Quality of the decomposition is in favor of *Singleshotpositive* for both *Movielens* and *Enron*. In addition, when the tensor size is small enough, *PosScalrandomtucker* is very computationally efficient, being one order of magnitude faster than our *Singleshot* approaches on *Enron*. However, *PosScalrandomtucker* is not able to decompose very large tensors and ran out of memory contrarily to *Singleshot*.

For illustrating the online capability of our algorithm, we have considered a tensor of size $20000 \times 2000 \times 50$ constructed from *Enron* which is artificially divided into slices drawn with respect to the first and the second modes. The core rank is $(R, R, R)$. We compare the online variant of our approach associated to positivity constraints named *Singleshotonlinepositive* to the online version of *Tensorsketch* [4] denoted *Tensorsketchonline*. Figure 3 shows running time for both algorithms. While of equivalent performance, our method is faster as our proposed update schemes, based on one single step of gradient descent, are more computationally efficient than a full alternate minimization.

**Remark 3.** *Other assessments are provided in the supplementary material: comparisons with other recent divide-and-conquer type approaches are provided, the non-nullity of the gradient with respect to $\mathbf{A}^{(n)}$ is numerically shown, and finally, we demonstrated the expected behavior of Singleshotinexact, i.e. "the higher the number of subtensors in the gradient approximation, the better performance we get".*

## 6 Conclusion

We have introduced two new algorithms named Singleshot and Singleshotinexact for scalable Tucker decomposition with convergence rates guarantees: for Singleshot, we have established a convergence rate to the set of minimizers of $\mathcal{O}(\frac{1}{\sqrt{K}})$ ($K$ being the maximum number of iterations) and for Singleshotinexact, a convergence rate of $\mathcal{O}\left(\frac{1}{k}\right)$ ($k$ being the iteration number). Besides, we have proposed a new approach for a problem that has drawn little attention so far, that is, the Tucker decomposition under the single pass constraint (with no need to resort to the past data) of a tensor streaming with respect to every mode. In future works, we aim at applying the principle of Singleshot to other decomposition problems different from Tucker.

## Acknowledgments

This work was supported by grants from the Normandie Projet GRR-DAISI, European funding FEDER DAISI and LEAUDS ANR-18-CE23-0020 Project of the French National Research Agency (ANR).

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
