[Supplementary Material]

# Singleshot : a scalable Tucker tensor decomposition Supplementary material

**Abraham Traoré**
LITIS EA4108
University of Rouen Normandy
abraham.traore@etu.univ-rouen.fr

**Maxime Bérar**
LITIS EA4108
University of Rouen Normandy
maxime.berar@univ-rouen.fr

**Alain Rakotomamonjy**
LITIS EA4108
University of Rouen Normandy
Criteo AI Lab, Criteo Paris
alain.rakoto@insa-rouen.fr

## 1 Properties on Tensors, Subtensors and Derivatives of the decomposition

### 1.1 Notations and definitions

Besides of the notations introduced in the main paper, we consider the following additional notations:

- For two tensors $\mathcal{X}, \mathcal{Y} \in \mathbb{R}^{I_1 \times \cdots \times I_N}$, we denote by $\langle, \rangle$ the element-wise inner product defined by $\langle \mathcal{X}, \mathcal{Y} \rangle = \sum_{i_1, \cdots, i_N} \mathcal{X}_{i_1, \cdots, i_N} \mathcal{Y}_{i_1, \ldots, i_N}$. This inner product is associated with the Frobenius norm defined by $\|\mathcal{X}\|_F^2 = \sum_{i_1, \cdots, i_N} \mathcal{X}_{i_1, \cdots, i_N}^2$.

- The Kronecker product of two matrices $\mathbf{A} \in \mathbb{R}^{K \times L}, \mathbf{B} \in \mathbb{R}^{M \times N}$, denoted by $\mathbf{A} \otimes \mathbf{B} \in \mathbb{R}^{KM \times LN}$ is defined by [6]:

$$\mathbf{A} \otimes B = \begin{bmatrix} \mathbf{A}_{1,1}\mathbf{B} \ldots . \mathbf{A}_{1,L}\mathbf{B} \\ \mathbf{A}_{2,1}\mathbf{B} \ldots . \mathbf{A}_{2,L}\mathbf{B} \\ \ldots \ldots \\ \ldots \ldots \\ \mathbf{A}_{K,1}B \ldots . \mathbf{A}_{K,L}\mathbf{B} \end{bmatrix}$$

- For $N$ matrices $\left\{ \mathbf{A}^{(n)} \right\}_1^N$, we denote the Kronecker product $\mathbf{A}^{(1)} \otimes \cdots \mathbf{A}^{(n-1)} \otimes \mathbf{A}^{(n+1)} \cdots \otimes \mathbf{A}^{(N)}$ by $\underset{m \in \mathbf{I}_{N \neq n}}{\otimes} \mathbf{A}^{(m)}$, $\otimes$ representing the Kronecker product of two matrices.

- The cardinality of a set $\theta$ is denoted by $\#\theta$.

- Let $\mathbf{A}^{(n)} \in \mathbb{R}^{I_n \times J_n}$ a matrix and $\theta$ a subset of $\{1, 2, ..., I_n\}$ (set of consecutive integers from 1 to $I_n$ with 1 and $I_n$ included). We denote by $\mathbf{A}_{\theta,:}^{(n)}$ the matrix derived from $\mathbf{A}^{(n)}$ by stacking row-wise the rows $\left\{ \mathbf{A}_{j,:}^{(n)}, j \in \theta \right\}$.

- For $\mathcal{X} \in \mathbb{R}^{I_1 \times \cdots \times I_N}$, we denote by $\mathcal{X}_{\theta_1}^1 \in \mathbb{R}^{\#\theta_1 \times I_2 \cdots \times I_N}$ a subtensor defined with respect to the first mode by fixing its first indexes to the values given by $\theta_1$, $\mathcal{X}_{\theta_1, \theta_2}^{1,2} \in \mathbb{R}^{\#\theta_1 \times \#\theta_2 \times I_3 \cdots \times I_N}$ a subtensor defined with respect to the first and second modes by fixing its first and second indexes to the values given by $\theta_1$ and $\theta_2$ and so on. For example, let's consider a four-order tensor $\mathcal{X} \in \mathbb{R}^{10 \times 15 \times 20 \times 30}$: $\mathcal{X}_{\theta_1}^1, \theta_1 = \{1, 6, 8, 9\}$ is the tensor derived from $\mathcal{X}$ by stacking with respect to the first mode the tensors $\left\{ \mathcal{X}_{e,:,:,:}^1, e \in \theta_1 \right\}$,

$\mathcal{X}^{1,2}_{\theta_1,\theta_2}, \theta_2 = \{1,2,4,5\}$ is the tensor such that $\left(\mathcal{X}^{1,2}_{\theta_1,\theta_2}\right)_{i,j,k,l} = \mathcal{X}_{i,j,k,l}$ with $(i,j) \in \theta_1 \times \theta_2$ and $\times$ being the Cartesian product of two sets.

- We denote $\mathrm{Tr}\,(\mathbf{A})$ the trace of a square matrix $\mathbf{A}$.

- Let's consider $N$ real numbers $a_1, .., a_N$. The products $a_1....a_N$ (product of the $N$ numbers) and $a_1..a_{n-1}a_{n+1}..a_N$ (product of all of the numbers except $a_n$) are denoted by:

$$a_1....a_N = \prod_{m \in \mathbf{I}_N} a_m$$

$$a_1..a_{n-1}a_{n+1}..a_N = \prod_{m \in \mathbf{I}_{N \neq n}} a_m$$

- The absolute value is denoted by $|\cdots|$

## 1.2 Properties on subtensors

The purpose of this section is to introduce some properties useful for the convergence results of *Singlesghot* and *Singleshot-inexact*. The first property expresses the mode-n subtensor of a tensor $\mathcal{X}$ depending on the latent factors which yield its Tucker decomposition. The reasoning uses simple algebraic arguments.

**Property 1** (Subtensor-Tucker decomposition). *Let $\mathcal{X} \in \mathbb{R}^{I_1 \times ... \times I_N}, \mathcal{G} \in \mathbb{R}^{J_1 \times ... \times J_N}$ be two tensors and $\left\{\mathbf{A}^{(m)} \in \mathbb{R}^{I_m \times J_m}, 1 \leq m \leq N\right\}$ N matrices such that:*

$$\mathcal{X} = \mathcal{G} \underset{m \in \mathbf{I}_N}{\times_m} \mathbf{A}^{(m)}$$

*The $n^{th}$ subtensor of $\mathcal{X}$ with respect to the mode $n$, denoted by $\mathcal{X}^n_{i_n} \in \mathbb{R}^{I_1 \times ... \times I_{n-1} \times 1 \times I_{n+1}..\times I_N}$, is equal to:*

$$\mathcal{X}^n_{i_n} = \mathcal{G} \underset{p \in \mathbf{I}_{n-1}}{\times_p} \mathbf{A}^{(p)} \times_n \mathbf{A}^{(n)}_{i_n,:} \underset{q \in \mathbf{I}^{n+1}_N}{\times_q} \mathbf{A}^{(q)}$$

*Besides, if $\theta \subset \{1,2,....I_n\}$ (set of consecutive integers from 1 to $I_n$ with 1 and $I_n$ included), we have:*

$$\mathcal{X}^n_\theta = \mathcal{G} \underset{p \in \mathbf{I}_{n-1}}{\times_p} \mathbf{A}^{(p)} \times_n \mathbf{A}^{(n)}_{\theta,:} \underset{q \in \mathbf{I}^{n+1}_N}{\times_q} \mathbf{A}^{(q)} \in \mathbb{R}^{I_1..\times I_{n-1} \times \#\theta \times I_{n+1}..\times I_N}, \# : cardinality$$

*Proof.* The mode-$n$ unfolding operator $[\cdot]^{(n)}$ turns a tensor in $\mathbb{R}^{I_1 \times \cdots \times I_N}$ into a matrix in $\mathbb{R}^{I_n \times \prod_{k \neq n} I_k}$ according to a given ordering. Let's denote $TZ_n$ its inverse operator that turns a matrix in $\mathbb{R}^{I_n \times \prod_{k \neq n} I_k}$ into a tensor in $\mathbb{R}^{I_1 \times \cdots \times I_N}$ according to the same ordering.
By definition of $\mathcal{X}^n_{i_n}$, we have:

$$\mathcal{X}^n_{i_n} = TZ_n\left(\mathbf{X}^{(n)}_{i_n,:}\right) \tag{1}$$

with $\mathbf{X}^{(n)}_{i_n,:}$ being the $i_n^{th}$ row of the matrix $\mathbf{X}^{(n)}$, the mode-n matricized form of the tensor $\mathcal{X}$.
By definition of the matricization of the Tucker decomposition, we have [7]:
$\mathbf{X}^{(n)} = \mathbf{A}^{(n)}\mathbf{G}^{(n)}\left(\otimes_{m \in \mathbf{I}_{N \neq n}} \mathbf{A}^{(m)}\right)^\top$
$\Rightarrow \mathbf{X}^{(n)}_{i_n,:} = \mathbf{A}^{(n)}_{i_n,:}\mathbf{G}^{(n)}\left(\otimes_{m \in \mathbf{I}_{N \neq n}} \mathbf{A}^{(m)}\right)^\top$ (because the rows of $\mathbf{AB}$ are defined by $\mathbf{A}_{i,:}\mathbf{B}$)
$\Rightarrow \mathbf{X}^{(n)}_{i_n,:} = \left[\mathcal{G} \underset{p \in \mathbf{I}_{n-1}}{\times_p} \mathbf{A}^{(p)} \times_n \mathbf{A}^{(n)}_{i_n,:} \underset{q \in \mathbf{I}^{n+1}_N}{\times_q} \mathbf{A}^{(q)}\right]^{(n)}$ (matricization of the Tucker decomposition [7])

The incorporation of the equality given by the last implication in the equation (1) yields:

$$\boldsymbol{\mathcal{X}}_{i_n}^n = TZ_n\left(\left[\boldsymbol{\mathcal{G}} \underset{p\in\mathbf{I}_{n-1}}{\times_p} \mathbf{A}^{(p)} \times_n \mathbf{A}_{i_n,:}^{(n)} \underset{q\in\mathbf{I}_N^{n+1}}{\times_q} \mathbf{A}^{(q)}\right]^{(n)}\right)$$

Given that $TZ_n$ is the inverse operator of $[\cdot]^{(n)}$ by definition, the last equality yields:

$$\boldsymbol{\mathcal{X}}_{i_n}^n = \boldsymbol{\mathcal{G}} \underset{p\in\mathbf{I}_{n-1}}{\times_p} \mathbf{A}^{(p)} \times_n \mathbf{A}_{i_n,:}^{(n)} \underset{q\in\mathbf{I}_N^{n+1}}{\times_q} \mathbf{A}^{(q)} \tag{2}$$

With the same reasoning, the second assertion is straightforward. $\square$

The next property expresses the objective function with respect to the subtensors drawn from $\boldsymbol{\mathcal{X}}$ and uses **Property** 1.

---

**Property 2** (Slice-wise expression of the Euclidean discrepancy). *Let $\boldsymbol{\mathcal{X}} \in \mathbb{R}^{I_1 \times \dots \times I_N}, \boldsymbol{\mathcal{G}} \in \mathbb{R}^{J_1 \times \dots \times J_N}$ two tensors, $\left\{\mathbf{A}^{(m)} \in \mathbb{R}^{I_m \times J_m}, 1 \leq m \leq N\right\}$ $N$ matrices. The following equality holds:*

$$\|\boldsymbol{\mathcal{X}} - \boldsymbol{\mathcal{G}} \underset{m\in\mathbf{I}_N}{\times_m} \mathbf{A}^{(m)}\|_F^2 = \sum_{i_n=1}^{I_n} \|\boldsymbol{\mathcal{X}}_{i_n}^n - \boldsymbol{\mathcal{G}} \underset{p\in\mathbf{I}_{n-1}}{\times_p} \mathbf{A}^{(p)} \times_n \mathbf{A}_{i_n,:}^{(n)} \underset{q\in\mathbf{I}_N^{n+1}}{\times_q} \mathbf{A}^{(q)}\|_F^2.$$

*Besides, if $\{\theta_m\}_{1 \leq m \leq M}$ is a partition of $\{1, 2, ..., I_n\}$, we have:*

$$\|\boldsymbol{\mathcal{X}} - \boldsymbol{\mathcal{G}} \underset{m\in\mathbf{I}_N}{\times_m} \mathbf{A}^{(m)}\|_F^2 = \sum_{m=1}^{M} \|\boldsymbol{\mathcal{X}}_{\theta_m}^n - \boldsymbol{\mathcal{G}} \underset{p\in\mathbf{I}_{n-1}}{\times_p} \mathbf{A}^{(p)} \times_n \mathbf{A}_{\theta_m,:}^{(n)} \underset{q\in\mathbf{I}_N^{n+1}}{\times_q} \mathbf{A}^{(q)}\|_F^2$$

*with $\boldsymbol{\mathcal{X}}_{\theta_m}^n \in \mathbb{R}^{I_1..I_{n-1} \times \#\theta_m \times I_{n+1} \times \dots \times I_N}$ and $\mathbf{A}_{\theta_m,:}^{(n)} \in \mathbb{R}^{\#\theta_m \times J_n}$.*

---

*Proof.* By definition of the square of the Frobenius norm, we have:
$\|\boldsymbol{\mathcal{X}} - \boldsymbol{\mathcal{G}} \underset{m\in\mathbf{I}_N}{\times_m} \mathbf{A}^{(m)}\|_F^2 = \sum_{i_n=1}^{I_n} \|\boldsymbol{\mathcal{X}}_{i_n}^n - \widehat{\boldsymbol{\mathcal{X}}}_{i_n}^n\|_F^2$ with $\widehat{\boldsymbol{\mathcal{X}}}_{i_n}^n$ being the $i_n^{th}$ subtensor of the tensor $\widehat{\boldsymbol{\mathcal{X}}} = \boldsymbol{\mathcal{G}} \underset{m\in\mathbf{I}_N}{\times_m} \mathbf{A}^{(m)}$ with respect to the mode $n$.

By **Property** 1, we have $\widehat{\boldsymbol{\mathcal{X}}}_{i_n}^n = \boldsymbol{\mathcal{G}} \underset{p\in\mathbf{I}_{n-1}}{\times_p} \mathbf{A}^{(p)} \times_n \mathbf{A}_{i_n,:}^{(n)} \underset{q\in\mathbf{I}_N^{n+1}}{\times_q} \mathbf{A}^{(q)}$, which concludes the proof for the first assertion.

The second assertion is straightforward with the second assertion of **Property** 1. $\square$

The next property simply states that the objective function can be expressed in terms of subtensors drawn with respect to every mode, a practical consequence being that we can choose them as small as we want.

---

**Property 3** (Expression of the Euclidean discrepancy with respect to subtensors drawn with respect to every mode). *Let's consider $\boldsymbol{\mathcal{X}} \in \mathbb{R}^{I_1 \times \dots \times I_N}$ a tensor and a partition $\left\{\theta_{m_n}^{(n)}, 1 \leq m_n \leq M_n\right\}$ a partition of $\{1, ..I_n\}$. Thus, we have:*
$f(\boldsymbol{\mathcal{G}}, \{\mathbf{A}^{(m)}\}_1^N) = \|\boldsymbol{\mathcal{X}} - \boldsymbol{\mathcal{G}} \underset{m\in\mathbf{I}_N}{\times_m} \mathbf{A}^{(m)}\|_F^2 = \sum_{m_1=1}^{M_1} .. \sum_{m_q=1}^{M_q} .. \sum_{m_N=1}^{M_N} \|\boldsymbol{\mathcal{X}}_{\theta_{m_1}^{(1)}, .., \theta_{m_N}^{(N)}} - \boldsymbol{\mathcal{G}} \underset{p\in\mathbf{I}_N}{\times_p}$
$\mathbf{A}_{\theta_{m_p}^{(p)},:}^{(p)}\|_F^2$ *with the subtensor $\boldsymbol{\mathcal{X}}_{\theta_{m_1}^{(1)}, .., \theta_{m_N}^{(N)}} \in \mathbb{R}^{\#\theta_{m_1}^{(1)} \times \dots \times \#\theta_{m_N}^{(N)}}$ being the subtensor derived from $\boldsymbol{\mathcal{X}}$ whose entries are $\boldsymbol{\mathcal{X}}_{i_1, \cdots, i_N}, (i_1, \cdots, i_N) \in \theta_{m_1}^{(1)} \times \cdots \times \theta_{m_N}^{(N)}, \mathbf{A}_{\theta_{m_p}^{(p)},:}^{(p)} \in \mathbb{R}^{\#\theta_{m_p}^{(p)} \times J_p}$*

---

*Proof.* By **Property** 1, we have:
$f\left(\boldsymbol{\mathcal{G}}, \{\mathbf{A}^{(m)}\}_1^N\right) = \sum_{m_1=1}^{M_1} \|\boldsymbol{\mathcal{X}}_{\theta_{m_1}^{(1)}}^1 - \boldsymbol{\mathcal{G}} \times_1 \mathbf{A}_{\theta_{m_1}^{(1)},:}^{(1)} \underset{m\in\mathbf{I}_N^2}{\times_m} \mathbf{A}^{(m)}\|_F^2.$

By applying **Property** 1 with respect to the second mode to $\|\boldsymbol{\mathcal{X}}^1_{\theta^{(1)}_{m_1}} - \boldsymbol{\mathcal{G}} \times_1 \mathbf{A}^{(1)}_{\theta^{(1)}_{m_1},:} \times_m \mathbf{A}^{(m)}\|^2_F$,

we have:
$$f\left(\boldsymbol{\mathcal{G}}, \{\mathbf{A}^{(m)}\}^N_1\right) = \sum_{m_1=1}^{M_1} \sum_{m_2=1}^{M_2} \|\boldsymbol{\mathcal{X}}^{1,2}_{\theta^{(1)}_{m_1},\theta^{(2)}_{m_2}} - \boldsymbol{\mathcal{G}} \times_1 \mathbf{A}^{(1)}_{\theta^{(1)}_{m_1},:} \times_2 \mathbf{A}^{(2)}_{\theta^{(2)}_{m_2},:} \times_m \mathbf{A}^{(m)}\|^2_F.$$

With the same recursive reasoning, we have:
$$f\left(\boldsymbol{\mathcal{G}}, \{\mathbf{A}^{(m)}\}^N_1\right) = \sum_{m_1=1}^{M_1} \sum_{m_2=1}^{M_2} \cdots \sum_{m_N=1}^{M_N} \|\boldsymbol{\mathcal{X}}_{\theta^{(1)}_{m_1},\ldots\theta^{(N)}_{m_N}} - \boldsymbol{\mathcal{G}} \times_{q\in I_N} \mathbf{A}^{(q)}_{\theta^{(q)}_{m_q},:}\|^2_F \qquad \square$$

## 1.3 Derivatives computation

This equality is going to be used to establish the expression of the derivative of our objective function with respect to the core and uses simple algebraic arguments.

---

**Property 4** (Equality on the element-wise inner product with a mode-n product). *Let's consider 2 tensors $\boldsymbol{\mathcal{X}} \in \mathbb{R}^{I_1\times\cdots\times I_N}, \boldsymbol{\mathcal{G}} \in \mathbb{R}^{J_1\times\cdots\times J_N}$ and N matrices $\left\{\mathbf{A}^{(m)}\right\}_{1\le m\le N}$. The following equality holds*

$$\langle \boldsymbol{\mathcal{X}}, \boldsymbol{\mathcal{G}} \underset{m\in\mathbf{I}_N}{\times_m} \mathbf{A}^{(m)}\rangle = \langle \boldsymbol{\mathcal{X}} \underset{m\in\mathbf{I}_N}{\times_m} \mathbf{A}^{(m)\top}, \boldsymbol{\mathcal{G}}\rangle$$

---

*Proof.* The tensorial element-wise inner product is by definition equivalent to the matricial element wise inner-product applied on any mode-n matricization of the two tensors. Thus, we have:

$\langle \boldsymbol{\mathcal{X}}, \boldsymbol{\mathcal{G}} \underset{m\in\mathbf{I}_N}{\times_m} \mathbf{A}^{(m)}\rangle = \langle \mathbf{X}^{(n)}, \mathbf{A}^{(n)}\mathbf{G}^{(n)} \left(\otimes_{m\in\mathbf{I}_{N\neq n}}\mathbf{A}^{(m)}\right)^\top\rangle$

$\Rightarrow \langle \boldsymbol{\mathcal{X}}, \boldsymbol{\mathcal{G}} \underset{m\in\mathbf{I}_N}{\times_m} \mathbf{A}^{(m)}\rangle = \text{Tr}\left(\mathbf{X}^{(n)\top}\mathbf{A}^{(n)}\mathbf{G}^{(n)} \left(\otimes_{m\in\mathbf{I}_{N\neq n}}\mathbf{A}^{(m)}\right)^\top\right)$ (by definition the inner product for the matrices)

$\Rightarrow \langle \boldsymbol{\mathcal{X}}, \boldsymbol{\mathcal{G}} \underset{m\in\mathbf{I}_N}{\times_m} \mathbf{A}^{(m)}\rangle = \text{Tr}\left(\left(\otimes_{m\in\mathbf{I}_{N\neq n}}\mathbf{A}^{(m)}\right)\mathbf{G}^{(n)\top}\mathbf{A}^{(n)\top}\mathbf{X}^{(n)}\right)$ (because $\text{Tr}(\mathbf{A}) = \text{Tr}(\mathbf{A}^T)$)

$\Rightarrow \langle \boldsymbol{\mathcal{X}}, \boldsymbol{\mathcal{G}} \underset{m\in\mathbf{I}_N}{\times_m} \mathbf{A}^{(m)}\rangle = \text{Tr}\left(\mathbf{G}^{(n)\top}\mathbf{A}^{(n)\top}\mathbf{X}^{(n)} \left(\otimes_{m\in\mathbf{I}_{N\neq n}}\mathbf{A}^{(m)}\right)\right)$ (because $\text{Tr}(\mathbf{AB}) = \text{Tr}(\mathbf{BA})$)

$\Rightarrow \langle \boldsymbol{\mathcal{X}}, \boldsymbol{\mathcal{G}} \underset{m\in\mathbf{I}_N}{\times_m} \mathbf{A}^{(m)}\rangle = \text{Tr}\left(\mathbf{G}^{(n)\top}\left[\boldsymbol{\mathcal{X}} \times_{m\in\mathbf{I}_N} \mathbf{A}^{(m)\top}\right]^{(n)}\right)$ (by definition of the matricization of the Tucker decomposition [7])

$\Rightarrow \langle \boldsymbol{\mathcal{X}}, \boldsymbol{\mathcal{G}} \underset{m\in\mathbf{I}_N}{\times_m} \mathbf{A}^{(m)}\rangle = \langle \mathbf{G}^{(n)}, \left[\boldsymbol{\mathcal{X}} \times_{m\in\mathbf{I}_N} \mathbf{A}^{(m)\top}\right]^{(n)}\rangle$ (by definition of the inner product for matrices)

$\Rightarrow \langle \boldsymbol{\mathcal{X}}, \boldsymbol{\mathcal{G}} \underset{m\in\mathbf{I}_N}{\times_m} \mathbf{A}^{(m)}\rangle = \langle \boldsymbol{\mathcal{X}} \underset{m\in\mathbf{I}_N}{\times_m} \mathbf{A}^{(m)\top}, \boldsymbol{\mathcal{G}}\rangle$

$\square$

As the previous one, this equality is going to be used to establish the expression of the derivative of our objective function with respect to the core and uses simple algebraic arguments. Mainly it uses the independance of two mode-$n$ products when the modes are different and the matricial product of factors with successive mode-$n$ products on the same mode.

---

**Property 5** (Successive mode-n products). *Let's consider a tensor $\boldsymbol{\mathcal{G}} \in \mathbb{R}^{J_1\times\cdots\times J_N}$ and N matrices $\left\{\mathbf{A}^{(m)} \in \mathbb{R}^{I_m\times J_m}, 1\le m\le N\right\}$. The following equality holds*

$$\left(\boldsymbol{\mathcal{G}} \underset{m\in\mathbf{I}_N}{\times_m} \mathbf{A}^{(m)}\right) \underset{m\in\mathbf{I}_N}{\times_m} \mathbf{A}^{(m)\top} = \boldsymbol{\mathcal{G}} \underset{m\in\mathbf{I}_N}{\times_m} \mathbf{A}^{(m)\top}\mathbf{A}^{(m)}$$

---

*Proof.* Among the known facts on mode-$n$ products [6], one is the independance regarding to the mode, for $m \neq n$, $\boldsymbol{\mathcal{X}} \times_n \mathbf{A} \times_m \mathbf{B} = \boldsymbol{\mathcal{X}} \times_m \mathbf{B} \times_n \mathbf{A}$, the other concerns successive mode-n products $\boldsymbol{\mathcal{X}} \times_n \mathbf{A} \times_n \mathbf{B} = \boldsymbol{\mathcal{X}} \times_n (\mathbf{BA})$ (provided the product $\mathbf{AB}$ makes sense).

$$\left(\boldsymbol{\mathcal{G}} \underset{m\in\mathbf{I}_N}{\times_m} \mathbf{A}^{(m)}\right) \underset{m\in\mathbf{I}_N}{\times_m} \mathbf{A}^{(m)\top} = \boldsymbol{\mathcal{G}} \underset{m\in\mathbf{I}_N}{\times_m} \mathbf{A}^{(m)} \underset{m\in\mathbf{I}_N}{\times_m} \mathbf{A}^{(m)\top} = \boldsymbol{\mathcal{G}} \underset{m\in\mathbf{I}_N}{\times_m} \mathbf{A}^{(m)\top}\mathbf{A}^{(m)}$$

□

The next property yields the expression of the derivative of the objective function with respect to the core as well as the Lipschitz character of the derivative under boundedness assumption and uses simple algebraic arguments

**Property 6** (Derivative/Lipschitz character of the derivative ). *The derivative of the function $\xi(\mathcal{G}) = \|\mathcal{X} - \mathcal{G} \underset{m \in \mathbf{I}_N}{\times_m} \mathbf{A}^{(m)}\|_F^2$ is given by*

$$\partial_{\mathcal{G}} \xi(\mathcal{G}) = -2 \left( \mathcal{X} - \mathcal{G} \underset{m \in \mathbf{I}_N}{\times_m} \mathbf{A}^{(m)} \right) \underset{m \in \mathbf{I}_N}{\times_m} \mathbf{A}^{(m)\top}.$$

*Let's assume that $\mathcal{G} \in \mathbb{D}_g$ and $\mathbf{A}^{(m)} \in \mathbb{D}_m, 1 \leq m \leq N$ with $\mathbb{D}_g = \{\mathcal{G} \in \mathbb{R}^{J_1 \times \ldots \times J_N} | \|\mathcal{G}\|_F \leq \alpha\}, \mathbb{D}_m = \{\mathbf{A}^{(m)} \in \mathbb{R}^{I_m \times J_m} | \|\mathbf{A}^{(m)}\|_F \leq \alpha\}$. Then, the derivative is Lipschitz with the bound $2 \prod_{m \in I_N} \|\mathbf{A}^{(m)}\|_F^2$*

*Proof.* Derivative and Lipschitz character.
1. Derivative expression justification
$\xi(\mathcal{G} + \mathcal{H}) = \|\mathcal{X} - (\mathcal{G} + \mathcal{H}) \underset{m \in \mathbf{I}_N}{\times_m} \mathbf{A}^{(m)}\|_F^2 = \|\mathcal{X} - \mathcal{G} \underset{m \in \mathbf{I}_N}{\times_m} \mathbf{A}^{(m)} - \mathcal{H} \underset{m \in \mathbf{I}_N}{\times_m} \mathbf{A}^{(m)}\|_F^2$
$\Rightarrow \xi(\mathcal{G} + \mathcal{H}) = \|\mathcal{R} - \mathcal{H} \underset{m \in \mathbf{I}_N}{\times_m} \mathbf{A}^{(m)}\|_F^2$ with $\mathcal{R} = \mathcal{X} - \mathcal{G} \underset{m \in \mathbf{I}_N}{\times_m} \mathbf{A}^{(m)}$ the residual tensor.
By definition of the square of the Frobenius norm, we have:
$\xi(\mathcal{G} + \mathcal{H}) = \xi(\mathcal{G}) - 2\langle \mathcal{R}, \mathcal{H} \underset{m \in \mathbf{I}_N}{\times_m} \mathbf{A}^{(m)} \rangle + O(\|\mathcal{H}\|_F^2)$
$\Rightarrow \xi(\mathcal{G} + \mathcal{H}) = \xi(\mathcal{G}) - 2\langle \mathcal{R} \underset{m \in \mathbf{I}_N}{\times_m} \mathbf{A}^{(m)\top}, \mathcal{H} \rangle + O(\|\mathcal{H}\|_F^2)$ (by **Property** 4).

The derivative is then $\partial_{\mathcal{G}} \xi(\mathcal{G}) = -2 \left( \mathcal{X} - \mathcal{G} \underset{m \in \mathbf{I}_N}{\times_m} \mathbf{A}^{(m)} \right) \underset{m \in \mathbf{I}_N}{\times_m} \mathbf{A}^{(m)\top}$
2. Justification of the Lipschitzian character
The derivative can be rewritten according to **Property** 5 as:
$\partial_{\mathcal{G}} \xi(\mathcal{G}) = -2\mathcal{X} \underset{m \in \mathbf{I}_N}{\times_m} \mathbf{A}^{(m)\top} + 2\mathcal{G} \underset{m \in \mathbf{I}_N}{\times_m} \mathbf{A}^{(m)\top} \mathbf{A}^{(m)}$,
The norm of the difference of the derivarives is $\|\partial_{\mathcal{G}} \xi(\mathcal{G}_1) - \partial_{\mathcal{G}} \xi(\mathcal{G}_2)\|_F = 2\|(\mathcal{G}_1 - \mathcal{G}_2) \underset{m \in \mathbf{I}_N}{\times_m}$
$\mathbf{A}^{(m)\top} \mathbf{A}^{(m)}\|_F$ and is bounded by:

$$\|\partial_{\mathcal{G}} \xi(\mathcal{G}_1) - \partial_{\mathcal{G}} \xi(\mathcal{G}_2)\|_F \leq 2\|\mathcal{G}_1 - \mathcal{G}_2\|_F \prod_{m \in \mathbf{I}_N} \|\mathbf{A}^{(m)\top} \mathbf{A}^{(m)}\|_F$$

$$\leq 2\|\mathcal{G}_1 - \mathcal{G}_2\|_F \prod_{m \in \mathbf{I}_N} \|\mathbf{A}^{(m)}\|_F^2$$

□

At this point, we simply derive the expression of the derivative of the objective function with respect to the subtensors

**Property 7** (Slice-wise derivative with respect to the core). *Let's consider the function $f$ defined by:*
$f(\mathcal{G}) = \sum_{i_n=1}^{I_n} \|\mathcal{X}_{i_n}^n - \mathcal{G} \underset{m \in \mathbf{I}_{n-1}}{\times_m} \mathbf{A}^{(m)} \times_n \mathbf{A}_{i_n,:}^{(n)} \underset{q \in \mathbf{I}_N^{n+1}}{\times_q} \mathbf{A}^{(q)}\|_F^2$. *The derivative of $f$ is given by:*

$$\partial_{\mathcal{G}} f(\mathcal{G}) = -2 \sum_{i_n=1}^{I_n} \left( \mathcal{X}_{i_n}^n - \mathcal{G} \underset{m \in \mathbf{I}_{N \neq n}}{\times_m} \mathbf{A}^{(m)} \times_n \mathbf{A}_{i_n,:}^{(n)} \right) \underset{m \in \mathbf{I}_{N \neq n}}{\times_m} \mathbf{A}^{(m)\top} \times_n \left( \mathbf{A}_{i_n,:}^{(n)} \right)^\top$$

*Proof.* This a direct consequence of **Property** 6. □

The purpose of the following property is to establish the expression of the derivative of the objective function with respect to a loading matrix as well as the Lipschitz character of the derivative under the boundedness assumption

**Property 8** (Derivative with respect loading matrix/ Lipschitz character). *Let's denote* $g(\mathbf{A}^{(n)}) = \|\mathcal{X} - \mathcal{G} \times_1 \mathbf{A}^{(1)} \times_2 \cdots \times_N \mathbf{A}^{(N)}\|_F^2$. *The derivative of* $g(\cdot)$ *is given by:*

$$\partial g(\mathbf{A}^{(n)}) = -2\left(\mathbf{X}^{(n)} - \mathbf{A}^{(n)}\mathbf{B}^{(n)}\right)\mathbf{B}^{(n)\top}$$

*with* $\mathbf{X}^{(n)}, \mathbf{G}^{(n)}, \mathbf{B}^{(n)}$ *being the mode-n matricized forms of the tensors* $\mathcal{X}, \mathcal{G}$, *and* $\mathcal{B}$ *defined as*

$$\mathcal{B} = \mathcal{G} \underset{p\in\mathbf{I}_{n-1}}{\times_p} \mathbf{A}^{(p)} \times_n \mathbf{Id} \underset{q\in\mathbf{I}_N^{n+1}}{\times_q} \mathbf{A}^{(q)},$$

*with* $\mathbf{Id} \in \mathbb{R}^{J_n \times J_n}$ *the identity matrix.*
*Let's assume that* $\mathcal{G} \in \mathbb{D}_g$ *and* $\mathbf{A}^{(m)} \in \mathbb{D}_m, 1 \leq m \leq N$ *with* $\mathbb{D}_g = \left\{\mathcal{G} \in \mathbb{R}^{J_1\times..\times J_N} |\|\mathcal{G}\|_F \leq \alpha\right\}, \mathbb{D}_m = \left\{\mathbf{A}^{(m)} \in \mathbb{R}^{I_m \times J_m} |\|\mathbf{A}^{(m)}\|_F \leq \alpha\right\}$. *The derivative* $\partial g$ *is Lipschitz with the bound* $2\|\mathcal{G}\|_F^2 \prod_{m\in I_{N\neq n}} \|\mathbf{A}^{(m)}\|_F^2$

*Proof.* :Derivative and Lipschitz character.
1. Derivative expression
By introducing the tensors $\mathcal{R} = \mathcal{X} - \mathcal{G} \underset{m\in\mathbf{I}_N}{\times_m} \mathbf{A}^{(m)}$ and $\mathcal{B} = \mathcal{G} \underset{p\in\mathbf{I}_{n-1}}{\times_p} \mathbf{A}^{(p)} \times_n \mathbf{Id} \underset{q\in\mathbf{I}_N^{n+1}}{\times_q} \mathbf{A}^{(q)}, \mathbf{Id} \in$
$\mathbb{R}^{J_n \times J_n}$ and by denoting $\mathbf{R}^{(n)}$ and $\mathbf{B}^{(n)}$ their mode-n matricized forms, we have:

$$g(\mathbf{A}^{(n)} + \mathbf{H}) = \|\mathcal{X} - \mathcal{G} \underset{m\in\mathbf{I}_N}{\times_m} \mathbf{A}^{(m)} - \mathcal{G} \underset{m\in\mathbf{I}_{N\neq n}}{\times_m} \mathbf{A}^{(m)} \times_n \mathbf{H}\|_F^2 = \|\mathcal{R} - \mathcal{B} \times_n \mathbf{H}\|_F^2$$

$$\Rightarrow g(\mathbf{A}^{(n)} + \mathbf{H}) = \|\mathcal{R}\|_F^2 - 2\left\langle \mathbf{R}^{(n)}, \mathbf{H}\mathbf{B}^{(n)}\right\rangle + \|\mathbf{H}\mathbf{B}^{(n)}\|_F^2$$

$$\Rightarrow g(\mathbf{A}^{(n)} + \mathbf{H}) = g(\mathbf{A}^{(n)}) - 2\left\langle \mathbf{R}^{(n)}\mathbf{B}^{(n)\top}, \mathbf{H}\right\rangle + \mathcal{O}\left(\|\mathbf{H}\|_F^2\right)$$

As a consequence and given that $\mathcal{G} \underset{m\in\mathbf{I}_N}{\times_m} \mathbf{A}^{(m)} = \mathcal{B} \times_n \mathbf{A}^{(n)}$, the derivative is:

$$\partial g(\mathbf{A}^{(n)}) = -2\mathbf{R}^{(n)}\mathbf{B}^{(n)\top} = -2\left(\mathbf{X}^{(n)} - \mathbf{A}^{(n)}\mathbf{B}^{(n)}\right)\mathbf{B}^{(n)\top}$$

2. Lipschitz character of the derivative
$\|\partial g(\mathbf{A}_1^{(n)}) - \partial g(\mathbf{A}_2^{(n)})\|_F = 2\|\left(\mathbf{A}_1^{(n)} - \mathbf{A}_2^{(n)}\right)\mathbf{B}^{(n)}\mathbf{B}^{(n)T}\|_F \leq 2\|\mathbf{A}_1^{(n)} - \mathbf{A}_2^{(n)}\|_F\|\mathbf{B}^{(n)}\|_F^2$
$\|\mathbf{B}^{(n)}\|_F = \|\mathbf{G}^{(n)} \underset{m\in\mathbf{I}_{N\neq n}}{\otimes} \mathbf{A}^{(m)}\|_F \leq \|\mathcal{G}\|_F \|\underset{m\in\mathbf{I}_{N\neq m}}{\otimes} \mathbf{A}^{(m)}\|_F = \|\mathcal{G}\|_F \prod_{m\in\mathbf{I}_{N\neq m}} \|\mathbf{A}^{(m)}\|_F$ (see for
example [8], page 433). Thus, we have:
$\|\partial g(\mathbf{A}_1^{(n)}) - \partial g(\mathbf{A}_2^{(n)})\|_F \leq 2\|\mathbf{A}_1^{(n)} - \mathbf{A}_2^{(n)}\|_F\|\mathcal{G}\|_F^2 \prod_{m\in\mathbf{I}_{N\neq m}} \|\mathbf{A}^{(m)}\|_F^2$ □

We simply derive the expression of the derivative of the objective function with respect to a loading matrix in terms of the subtensors

**Property 9** (Slice-wise derivative with respect to a loading matrix). *Let's denote* $f(\mathbf{A}^{(p)}) = \sum_{i_n=1}^{I_n} \|\boldsymbol{\mathcal{X}}_{i_n}^n - \boldsymbol{\mathcal{G}} \underset{p \in \mathbf{I}_{n-1}}{\times_p} \mathbf{A}^{(p)} \times_n \mathbf{A}_{i_n,:}^{(n)} \underset{q \in \mathbf{I}_N^{n+1}}{\times_q} \mathbf{A}^{(q)}\|_F^2$ *with* $p < n$ *($p > n$ follows the same principle). The derivative of* $f$ *is given by*

$$\partial f(\mathbf{A}^{(p)}) = -2 \sum_{i_n=1}^{I_n} \left( \left(\mathbf{X}_{i_n}^n\right)^{(p)} - \mathbf{A}^{(p)} \left(\mathbf{B}_{i_n}^{(p)}\right)\right)\left(\mathbf{B}_{i_n}^{(p)}\right)^\top$$

*with* $\left(\mathbf{X}_{i_n}^n\right)^{(p)}$ *and* $\mathbf{B}_{i_n}^{(p)}$ *being the mode-p matricized forms of the subtensors* $\boldsymbol{\mathcal{X}}_{i_n}^n$ *and* $\boldsymbol{\mathcal{B}}_{i_n}$, *with* $\boldsymbol{\mathcal{B}}_{i_n}$ *defined by*

$$\boldsymbol{\mathcal{B}}_{i_n} = \boldsymbol{\mathcal{G}} \underset{m \in \mathbf{I}_{p-1}}{\times_m} \mathbf{A}^{(m)} \times_p \mathbf{Id} \underset{q \in \mathbf{I}_{n-1}^{p+1}}{\times_q} \mathbf{A}^{(q)} \times_n \mathbf{A}_{i_n,:}^{(n)} \underset{r \in \mathbf{I}_N^{n+1}}{\times_r} \mathbf{A}^{(r)}.$$

*Proof.* This is a direct application of **Property** 8. □

The following inequality is useful for the proof of *Singleshot-inexact*

**Property 10.** *For* $\lambda > 0$ *and* $k > k_0 = 1 + \frac{1}{\log(1+\lambda)}\log\left(\frac{1}{\log(1+\lambda)}\right)$, $\frac{1}{(1+\lambda)^{k-1}} \leq \frac{1}{k-k_0}$ *with* $\log$ *being the logarithmic function, i.e. the inverse of the exponential function which we denote by* $e$.

*Proof.* let's consider the univariate function $\ell(x) = x - e^{(x-1)\log(1+\lambda)} - k_0$ defined on the domain $x > k_0$, $e$ being the exponential function (inverse of the logarithmic function).
The derivative is given by $\ell'(x) = 1 - (\log(1+\lambda))e^{(x-1)\times ln(1+\lambda)}$.
Given that $x > k_0$, we have by definition of $k_0$:
$x > 1 + \frac{1}{\log(1+\lambda)}\log(\frac{1}{\log(1+\lambda)})$
$\Rightarrow x - 1 > \frac{1}{\log(1+\lambda)}\log(\frac{1}{\log(1+\lambda)})$
$\Rightarrow (x-1)\log(1+\lambda) \geq \log\left(\frac{1}{\log(1+\lambda)}\right)$ (because $\log(1+\lambda) > 0$ since $\lambda > 0$ by definition)
$\Rightarrow e^{(x-1)\log(1+\lambda)} \geq e^{\log\left(\frac{1}{\log(1+\lambda)}\right)}$ (because the exponential $e$ is an increasing function)
$\Rightarrow e^{(x-1)\log(1+\lambda)} \geq \frac{1}{\log(1+\lambda)}$ (because $e$ is the inverse of the logarithmic function)
$\Rightarrow e^{(x-1)\log(1+\lambda)}\log(1+\lambda) \geq 1$ (because $\log(1+\lambda) > 0$ since $\lambda > 0$ by definition)
$\Rightarrow 0 \geq 1 - (\log(1+\lambda))e^{(x-1)\log(1+\lambda)}$
$\Rightarrow \ell'(x) \leq 0$: this implies that $\ell$ is a decreasing function on the domain $x > k_0$.
Thus, for an integer $k > k_0$, we have:
$\ell(k) = k - e^{(k-1)\log(1+\lambda)} - k_0 \leq \ell(k_0) = -e^{(k_0-1)\log(1+\lambda)} < 0$
$\Rightarrow 0 < k - k_0 < e^{(k-1)\log(1+\lambda)} = (1+\lambda)^{k-1}$: this concludes the proof. □

## 2 Theoretical analysis

### 2.1 Definitions and supplementary notations

$$f\left(\boldsymbol{\mathcal{G}}, \mathbf{A}^{(1)}, \cdots, \mathbf{A}^{(N)}\right) = \frac{1}{2}\|\boldsymbol{\mathcal{X}} - \boldsymbol{\mathcal{G}} \times_{m \in I_N} \mathbf{A}^{(m)}\|_F^2$$

$$= \frac{1}{2}\sum_{i_n=1}^{I_n} \|\boldsymbol{\mathcal{X}}_{i_n}^n - \boldsymbol{\mathcal{G}} \underset{p \in \mathbf{I}_{n-1}}{\times_p} \mathbf{A}^{(p)} \times_n \mathbf{A}_{i_n,:}^{(n)} \underset{q \in I_N^{n+1}}{\times_q} \mathbf{A}^{(q)}\|_F^2$$

We consider, for writing simplicity, the alternative notations of $f\left(\boldsymbol{\mathcal{G}}, \mathbf{A}^{(1)}, \cdots, \mathbf{A}^{(N)}\right)$ given by $f\left(\boldsymbol{\mathcal{G}}, \{\mathbf{A}^{(m)}\}_1^N\right)$, $f\left(\boldsymbol{\mathcal{G}}, \{\mathbf{A}^{(p)}\}_1^n, \{\mathbf{A}^{(q)}\}_{n+1}^N\right)$, $f\left(\boldsymbol{\mathcal{G}}, \{\mathbf{A}^{(p)}\}_1^{n-1}, \mathbf{A}^{(n)}, \{\mathbf{A}^{(q)}\}_{n+1}^N\right)$ and the same notations hold for any function of the $N+1$ variables $\{\boldsymbol{\mathcal{G}}, \mathbf{A}^{(1)}, \cdots, \mathbf{A}^{(N)}\}$.
Besides, we consider the following notations:

- the minimum value of $f$ by $f_{\min} = f\left(\boldsymbol{\mathcal{G}}_m, \mathbf{A}_m^{(1)}, .., \mathbf{A}_m^{(N)}\right)$, with

$$\left(\boldsymbol{\mathcal{G}}_m, \mathbf{A}_m^{(1)}, \cdots, \mathbf{A}_m^{(N)}\right) \leftarrow \underset{\mathbb{D}_g \times \mathbb{D}_1 \times .... \times \mathbb{D}_N}{\arg\min} f\left(\boldsymbol{\mathcal{G}}, \mathbf{A}^{(1)}, \cdots, \mathbf{A}^{(N)}\right).$$

Such value exists since a finite product of compact sets is a compact set and $f$ is continuous since it is polynomial.

- the function $f$ considered as a function of only the variable $\boldsymbol{\mathcal{G}}$ by $f(., \{\mathbf{A}^{(m)}\}_1^N)$. The same notations hold for all the variables as well as the derivatives.

- the block-wise derivative of $f$ with respect to $\boldsymbol{\mathcal{G}}$ (respectively with respect to $\mathbf{A}^{(p)}$) evaluated at $\left(\tilde{\boldsymbol{\mathcal{G}}}, \tilde{\mathbf{A}}^{(1)}, \cdots, \tilde{\mathbf{A}}^{(N)}\right)$ by $\partial_{\boldsymbol{\mathcal{G}}} f\left(\tilde{\boldsymbol{\mathcal{G}}}, \left\{\tilde{\mathbf{A}}^{(m)}\right\}_1^N\right)$ (respectively $\partial_{\mathbf{A}^{(p)}} f\left(\tilde{\boldsymbol{\mathcal{G}}}, \left\{\tilde{\mathbf{A}}^{(m)}\right\}_1^N\right)$).

- The block-wise derivative of $f$ is denoted by $\partial_x f$ ($x$ representing the right variable). The gradient of $f$, denoted by $\nabla f = (\partial_{\boldsymbol{\mathcal{G}}} f, \partial_{\mathbf{A}^{(1)}} f, ..., \partial_{\mathbf{A}^{(N)}} f)$. is an element of $\mathbb{R}^{J_1 \times \cdots \times J_N} \times \mathbb{R}^{I_1 \times J_1} \times \cdots \times \mathbb{R}^{I_N \times J_N}$ endowed with the norm $\|\cdot\|_*$ defined as the sum of the Frobenius norms, i.e.:

$$\|\nabla f(\tilde{\boldsymbol{\mathcal{G}}}, \left\{\tilde{\mathbf{A}}^{(m)}\right\}_1^N)\|_* = \|\partial_{\boldsymbol{\mathcal{G}}} f\left(\tilde{\boldsymbol{\mathcal{G}}}, \left\{\tilde{\mathbf{A}}^{(m)}\right\}\right)\|_F + \sum_{p=1}^N \|\partial_{\mathbf{A}^{(p)}} f\left(\tilde{\boldsymbol{\mathcal{G}}}, \left\{\tilde{\mathbf{A}}^{(m)}\right\}\right)\|_F$$

We define the following suprema, which are well defined since $f$ and its derivatives are continuous (because polynomial) and a finite product of compact sets is a compact set :

- $\Gamma = \sup_{\mathbb{D}_g \times \mathbb{D}_1 \times \cdots \times \mathbb{D}_N} f\left(\boldsymbol{\mathcal{G}}, \mathbf{A}^{(1)}, \cdots, \mathbf{A}^{(N)}\right)$

- $\Gamma_g = \sup_{\mathbb{D}_g \times \mathbb{D}_1 \times .. \times \mathbb{D}_N} \|\partial_{\boldsymbol{\mathcal{G}}} f\left(\boldsymbol{\mathcal{G}}, \mathbf{A}^{(1)}, ..., \mathbf{A}^{(N)}\right)\|_F$

- $\Gamma_m = \sup_{\mathbb{D}_g \times \mathbb{D}_1 \times ... \times \mathbb{D}_N} \|\partial_{\mathbf{A}^{(m)}} f\left(\boldsymbol{\mathcal{G}}, \mathbf{A}^{(1)}, ..., \mathbf{A}^{(N)}\right)\|_F$

Lastly, the maximum number of iterations will be denoted by $K$

### 2.1.1 Definitions for *Singleshot-inexact*

We consider the following definition for $\eta_k^N$ (descent step for $\mathbf{A}^{(N)}$ at the iteration $k+1$)

$$\eta_k^N = \underset{\eta \in [\frac{1}{4K\gamma}, \frac{1}{K\gamma}]}{\arg\min} \sigma(\eta) \left(\lambda f\left(\boldsymbol{\mathcal{G}}_{k+1}, \left\{\mathbf{A}_{k+1}^{(m)}\right\}_1^{N-1}, \tilde{\mathbf{A}}^{(N)}(\eta)\right) - f\left(\boldsymbol{\mathcal{G}}_k, \left\{\mathbf{A}_k^{(m)}\right\}_1^N\right) + f\left(\boldsymbol{\mathcal{G}}_{k+1}, \left\{\mathbf{A}_{k+1}^{(m)}\right\}_1^{N-1}, \mathbf{A}_k^{(N)}\right)\right)$$

(3)

with:

$$\sigma(\eta) = \eta - \frac{1}{4K\gamma}$$

$$\ell_j(\mathbf{A}^{(N)}) = \frac{1}{2}\|\boldsymbol{\mathcal{X}}_j^n - \boldsymbol{\mathcal{G}}_{k+1} \underset{m \in I_{n-1}}{\times_m} \mathbf{A}_{k+1}^{(m)} \times_n (\mathbf{A}_{k+1}^{(n)})_{j,:} \underset{q \in I_{N-1}^{n+1}}{\times_q} \mathbf{A}_{k+1}^{(q)} \times_N \mathbf{A}^{(N)}\|_F^2$$

$$\tilde{\mathbf{A}}^{(N)}(\eta) = \mathbf{A}_k^{(N)} - \eta \times \frac{1}{B_k} \sum_{j \in \mathcal{SET}_k} \partial_{\mathbf{A}^{(N)}} \ell_j$$

where $\partial_{\mathbf{A}^{(N)}} \ell_j$ represents the derivative of $\ell_j$ evaluated at $\mathbf{A}_k^{(N)}$, and $\lambda > 0, \gamma > 1$ being user-defined parameters.

This problem is well defined for two main reasons. The first one is that all the factors $\left\{\boldsymbol{\mathcal{G}}_k, \left\{\mathbf{A}_k^{(m)}\right\}_{1 \leq m \leq N}, \boldsymbol{\mathcal{G}}_{k+1}, \left\{\mathbf{A}_{k+1}^{(m)}\right\}_{1 \leq m \leq N-1}\right\}$ are known at the update stage of $\mathbf{A}^{(N)}$ at the $(k+1)^{th}$ iteration (i.e. the computation stage of $\mathbf{A}_{k+1}^{(N)}$) because the variables updates are performed in the order 'update of $\boldsymbol{\mathcal{G}}$', 'update of $\mathbf{A}^{(1)}$',..., 'update of $\mathbf{A}^{(N-1)}$', 'update of $\mathbf{A}^{(N)}$' and we consider the minimization problem of a continuous function on a compact set.

The second is that as a consequence of the **Assumption 3.2** presented below, there exists $\mathcal{SET}_k$ such that $\sum_{j \in \mathcal{SET}_k} \partial_{\mathbf{A}^{(N)}} \ell_j \neq 0$ (see **Property** 11) and the set considered is one of such sets (i.e. for which the inexact gradient is different from the null matrix). In the sequel, the matrix $\sum_{j \in \mathcal{SET}_k} \partial_{\mathbf{A}^{(N)}} \ell_j$ will be referred to as the **inexact gradient**.

### 2.1.2 Definitions for *Singleshot*

For *Singleshot*, we consider the following definitions for the descent steps $\eta_k^{\mathcal{G}}$ and $\eta_k^p$ at the $(k+1)^{th}$ iteration:

$$\eta_k^{\mathcal{G}} = \underset{\eta \in [\frac{\delta_1}{\sqrt{K}}, \frac{\delta_2}{\sqrt{K}}]}{\arg\min} \sigma(\eta) \left( f\left( \mathcal{G}_k - \eta \mathcal{D}_k^{\mathcal{G}}, \left\{ \mathbf{A}_k^{(m)} \right\}_1^N \right) - f\left( \mathcal{G}_k, \left\{ \mathbf{A}_k^{(m)} \right\}_1^N \right) \right) \qquad (4)$$

$$\eta_k^p = \underset{\eta \in [\frac{\delta_1}{\sqrt{K}}, \frac{\delta_2}{\sqrt{K}}]}{\arg\min} \sigma(\eta) \left( f\left( \mathcal{G}_{k+1}, \left\{ \mathbf{A}_{k+1}^{(m)} \right\}_1^{p-1}, \mathbf{A}_k^{(p)} - \eta \mathbf{D}_k^p, \left\{ \mathbf{A}_k^{(q)} \right\}_{p+1}^N \right) - f\left( \mathcal{G}_{k+1}, \left\{ \mathbf{A}_{k+1}^{(m)} \right\}_1^{p-1}, \left\{ \mathbf{A}_k^{(q)} \right\}_p^N \right) \right)$$
$$(5)$$

with:

$$\sigma(\eta) = \eta - \frac{\delta_1}{\sqrt{K}}$$

The parameters $\delta_1 > 0, \delta_2 > 0$ being user-defined parameters.

The minimization problems related to $\eta_k^{\mathcal{G}}$ and $\eta_k^p$ are well defined since the factors involved in their definitions are known at the computation stage of $\mathcal{G}_{k+1}$ and $\mathbf{A}_{k+1}^{(p)}$ and the problems correspond to the minimization of functions on compact sets.

**Remark 1.** *Contrary to* Singleshot, *only the descent step $\eta_k^N$ for the loading matrix $\mathbf{A}^{(N)}$ is defined via a minimization problem for* Singleshot-inexact

## 2.2 Assumptions

The theoretical analysis of *Singleshotinexact* requires **Assumption 1**, **Assumption 2**, **Assumption 3.1**, **Assumption 3.2**, **Assumption 3.3** and the definition of the step given by the equation (3). The theoretical analysis of *Singleshot* requires the **Assumption 1**, **Assumption 2**, **Assumption 4** and the minimization problems defined by the equations (4) and (5). All of the assumptions are given below.

### 2.2.1 Common assumptions

**Assumption 1** (Uniform boundedness). *The $n^{th}$ subtensors are uniformly bounded, i.e. $\|\mathcal{X}_{i_n}^n\|_F \leq \rho, \forall 1 \leq i_n \leq I_n$*

**Assumption 2** (Factors boundedness). *$\mathcal{G} \in \mathbb{D}_g = \left\{ \mathcal{G} \in \mathbb{R}^{J_1 \times .. \times J_N} | \|\mathcal{G}\|_F \leq \alpha \right\}, \mathbf{A}^{(m)} \in \mathbb{D}_m = \left\{ \mathbf{A}^{(m)} \in \mathbb{R}^{I_m \times J_m} | \|\mathbf{A}^{(m)}\|_F \leq \alpha \right\}, 1 \leq m \leq N$*

### 2.2.2 specific assumptions for *Singleshotinexact*

Besides of **Assumption 1** and **Assumption 2**, we consider three additional assumptions:

**Assumption 3.1** (Bound on descent step related to the update of $\mathbf{A}^{(N)}$).

$$\frac{1}{4K^\gamma} < \eta_k^N \leq \frac{1}{\alpha^{2N}}$$

**We highlight the fact that $\eta_k^N$ is does "not" mean "$\eta_k$ to the power $N$": $N$ simply refers to the loading matrix $\mathbf{A}^{(N)}$. On the other side, $\alpha^{2N}$ corresponds to $\alpha$ to the power $2N$.**

**Assumption 3.2** (non-nullity of the gradient with respect to $\mathbf{A}^{(N)}$).

$$\partial_{\mathbf{A}^{(N)}} f\left( \mathcal{G}_{k+1}, \left\{ \mathbf{A}_{k+1}^{(m)} \right\}_1^{N-1}, \mathbf{A}_k^{(N)} \right) \neq 0$$

*It is worth to notice that we do not impose the non-vanishing assumption on all of the partial gradients, but only on the partial gradient with respect to $\mathbf{A}^{(N)}$ evaluated at the point $\left( \mathcal{G}_{k+1}, \mathbf{A}_{k+1}^{(1)}, .., \mathbf{A}_{k+1}^{(N-1)}, \mathbf{A}_k^{(N)} \right)$*

**Assumption 3.3** (Choice of the number of subtensors).

$$I_n \sqrt{\frac{1}{2} + \frac{1}{I_n}} \leq B_k, I_n > 2$$

### 2.2.3 specific assumptions for *Singleshot*

Besides of the definitions given by the equations (4) and (5), we consider, alongside **Assumption 1** and **Assumption 2**, the following additional assumption:

**Assumption 4.**

$$\frac{\delta_1}{\sqrt{K}} < \eta_k^{\boldsymbol{\mathcal{G}}}, \eta_k^p \leq \frac{\delta_2}{\sqrt{K}}$$

These inequalities simply amount to consider that the solutions of the minimization problems given by the equations (4) and (5) are not attained at the lower bound of the interval. **Again, we highlight the fact that $\eta_k^p$ is does "not" mean "$\eta_k$ to the power $p$": $p$ simply refers to the loading matrix $\mathbf{A}^{(p)}$.**

## 2.3 Theoretical result

The theoretical analysis is motivated by the fact that the existing convergence results for *Coordinate Gradient Descent* cannot be applied directly to our setting. Most of the existing approaches use the assumption of convexity [1] or strong convexity [9] on the objective function, which is not verified in our case since we are in a non-convex setting. Some approaches have been proposed for the non-convex setting, but with block-wise (i.e. multivariate function considered as a function of one of its variable while the others are fixed) strong convexity [13]. The reasoning proposed does not fit our framework since the block-wise strong convexity is not verified for our problem.

Few approaches (in the sense that the problem has drawn much less attention compared to the convexity or strong convexity cases) handle, for any function, the general case with no convexity or block-wise convexity assumption (our problem can be classified in this trend), but the theoretical analysis cannot be replicated since the algorithmic frameworks are different from ours. Our purpose here is not to compete with state-of-the-art *Coordinate Gradient Descent* algorithms, but simply to prove such a simple algorithmic setting yields a convergence rate in the general non-convex setting for the scalable tensor decomposition problem.

The difficulty of the proof relies on the lack of convexity for the objective function as well as the block-wise function. To overcome this, we rely on careful definitions of the descent steps.

### 2.3.1 Convergence of *Singleshotinexact*

The purpose of this section is to prove that the sequence $\left\{\boldsymbol{\mathcal{G}}_k, \mathbf{A}_k^{(1)}, \cdots, \mathbf{A}_k^{(N)}\right\}$ converges to the set of minimizers of $f$ at the rate $\mathcal{O}(\frac{1}{k})$. Before establishing the convergence, we introduce the preliminary results **Property** 11, **Property** 12 and **Property** 13.

The aim of the following property is to prove that there exists a non-zero inexact gradient. It is simply based on a reasoning by contradiction and is a direct consequence of **Assumption 3.2**.

**Property 11** (Existence of a set such that the inexact gradient is non-zero). *Under **Assumption 3.2**, the following property holds:*
$\exists \mathcal{SET}_k \subset \{1, 2, \ldots I_n\}, \sum_{j \in \mathcal{SET}_k} \partial_{\mathbf{A}^{(N)}} \ell_j \neq 0$ *with $\partial_{\mathbf{A}^{(N)}} \ell_j$ defined by the equation* (3)

*Proof.* We perform a reasoning by contradiction. Let's assume that $\forall \mathcal{SET}_k \subset \{1, 2, \ldots, I_n\}, \sum_{j \in \mathcal{SET}_k} \partial_{\mathbf{A}^{(N)}} \ell_j = 0$
Let's choose $\mathcal{SET}_k = \{1, 2, \ldots, I_n\}$. Thus, we have

$$\sum_{j=1}^{I_n} \partial_{\mathbf{A}^{(N)}} \ell_j = \partial_{\mathbf{A}^{(N)}} f \left(\boldsymbol{\mathcal{G}}_{k+1}, \left\{\mathbf{A}_{k+1}^{(m)}\right\}_1^{N-1}, \mathbf{A}_k^{(N)}\right) = 0,$$

which is in contradiction with **Assumption 3.2** $\qquad\square$

For this part of the proof, we simply establish an explicit bound on the norm of the gradient approximation error. This idea that consists to bound the gradient approximation error $\|\bar{\beta}_k\|_F^2$ by

considering a lower bound on the number of terms that intervene in the inexact-gradient is inspired from [12].

---

**Property 12** (Bound on the gradient approximation error). *Let's consider the gradient approximation error $\beta_k$ defined by:*

$\beta_k = \frac{1}{B_k} \sum_{j \in \mathcal{SET}_k} \partial_{\mathbf{A}^{(N)}} \ell_j - \partial_{\mathbf{A}^{(N)}} f(\mathcal{G}_{k+1}, \left\{ \mathbf{A}_{k+1}^{(m)} \right\}_1^{N-1}, \mathbf{A}_k^{(N)})$ *with $\ell_j$ and $\partial_{\mathbf{A}^{(N)}} \ell_j$ defined by (3). The following inequality holds:*

$$\|\beta_k\|_F^2 \leq 8 I_n^2 (\rho + \alpha^{N+1})^2 \alpha^{2N}$$

---

*Proof.* By definition of $\beta_k$, we have:

$\|\beta_k\|_F^2 = \|\frac{1}{B_k} \sum_{j \in \mathcal{SET}_k} \partial_{\mathbf{A}^{(N)}} \ell_j - \partial_{\mathbf{A}^{(N)}} f(\mathcal{G}_{k+1}, \left\{ \mathbf{A}_{k+1}^{(m)} \right\}_1^{N-1}, \mathbf{A}_k^{(N)})\|_F^2$

Given that $\partial_{\mathbf{A}^{(N)}} f(\mathcal{G}_{k+1}, \left\{ \mathbf{A}_{k+1}^{(m)} \right\}_1^{N-1}, \mathbf{A}_k^{(N)}) = \sum_{j \in \mathcal{SET}_k} \partial_{\mathbf{A}^{(N)}} \ell_j + \sum_{j \notin \mathcal{SET}_k} \partial_{\mathbf{A}^{(N)}} \ell_j$ by the alternative expression of $f$ in terms of subtensors drawn with respect to one mode, we have:

$\|\beta_k\|_F^2 = \|\left( \frac{1}{B_k} - 1 \right) \sum_{j \in \mathcal{SET}_k} \partial_{\mathbf{A}^{(N)}} \ell_j - \sum_{j \notin \mathcal{SET}_k} \partial_{\mathbf{A}^{(N)}} \ell_j\|_F^2$

$\Rightarrow \|\beta_k\|_F^2 \leq \left( \left( 1 - \frac{1}{B_k} \right) \sum_{j \in \mathcal{SET}_k} \|\partial_{\mathbf{A}^{(N)}} \ell_j\|_F + \sum_{j \notin \mathcal{SET}_k} \|\partial_{\mathbf{A}^{(N)}} \ell_j\|_F \right)^2$ (since the real function $x \to x^2$ is increasing on the set of real positive numbers and by triangle inequality)

Since $(a+b)^2 \leq 2(a^2 + b^2)$ for $a, b \geq 0$, by application of Cauchy-Schwartz inequality and given that the cardinality of $\mathcal{SET}_k$ is $B_k$, we have:

$\|\beta_k\|_F^2 \leq 2 \left( 1 - \frac{1}{B_k} \right)^2 B_k \sum_{j \in \mathcal{SET}_k} \|\partial_{\mathbf{A}^{(N)}} \ell_j\|_F^2 + 2(I_n - B_k) \sum_{j \notin \mathcal{SET}_k} \|\partial_{\mathbf{A}^{(N)}} \ell_j\|_F^2$

Given that $\sum_{j \in \mathcal{SET}_k} = \sum_{1 \leq j \leq I_n} - \sum_{j \notin \mathcal{SET}_k}$, we have:

$$\|\beta_k\|_F^2 \leq 2 \left( 1 - \frac{1}{B_k} \right)^2 B_k \sum_{j=1}^{I_n} \|\partial_{\mathbf{A}^{(N)}} \ell_j\|_F^2 + 2 \underbrace{\left( (I_n - B_k) - \frac{(B_k - 1)^2}{B_k} \right)}_{\theta} \sum_{j \notin \mathcal{SET}_k} \|\partial_{\mathbf{A}^{(N)}} \ell_j\|_F^2$$

with $\theta = \frac{I_n B_k - B_k^2 - (B_k^2 - 2B_k + 1)}{B_k} \leq \frac{I_n B_k - 2B_k^2 + 2B_k - 1}{B_k} \leq \frac{I_n^2 - 2I_n^2(\frac{1}{2} + \frac{1}{I_n}) + 2I_n - 1}{B_k} = -\frac{1}{B_k}$ under **Assumption 3.3**.

Thus, by **Assumption 1**, **Assumption 2** and given that $B_k \leq I_n$, we have:

$\|\beta_k\|_F^2 \leq 2 \left( 1 - \frac{1}{B_k} \right)^2 B_k \sum_{j=1}^{I_n} \|\partial_{\mathbf{A}^{(N)}} \ell_j\|_F^2 \leq 2 \times 4 \times I_n \times I_n (\rho + \alpha^{N+1})^2 \alpha^{2N}$

$\Rightarrow \|\beta_k\|_F^2 \leq 8 I_n^2 (\rho + \alpha^{N+1})^2 \alpha^{2N}$ □

The following inequality simply establishes a lower bound on the variation of the block-wise function $\mathbf{A}^{(N)} \to f(\mathcal{G}_k, \left\{ \mathbf{A}_{k+1}^{(m)} \right\}_1^{N-1}, \mathbf{A}^{(N)})$ between the consecutive iterations $k$ and $k+1$ and is a direct consequence of the minimization problem given by the equation (3).

---

**Property 13** (Lower bound on the variation of the block-wise objective function). *The following inequality holds:*

$$\lambda f \left( \mathcal{G}_{k+1}, \left\{ \mathbf{A}_{k+1}^{(m)} \right\}_1^N \right) - f \left( \mathcal{G}_k, \left\{ \mathbf{A}_k^{(m)} \right\}_1^N \right) + f \left( \mathcal{G}_{k+1}, \left\{ \mathbf{A}_{k+1}^{(m)} \right\}_1^{N-1}, \mathbf{A}_k^{(N)} \right) \leq 0$$

---

*Proof.* The minimization problem defining $\eta_k^N$ being well defined (see the equation (3) and **Property** 11), we have:

$\sigma \left( \eta_k^N \right) \left( \lambda f \left( \mathcal{G}_{k+1}, \left\{ \mathbf{A}_{k+1}^{(m)} \right\}_1^{N-1}, \tilde{\mathbf{A}}^{(N)}(\eta_k^N) \right) - f \left( \mathcal{G}_k, \left\{ \mathbf{A}_k^{(m)} \right\}_1^N \right) + f \left( \mathcal{G}_{k+1}, \left\{ \mathbf{A}_{k+1}^{(m)} \right\}_1^{N-1}, \mathbf{A}_k^{(N)} \right) \right)$

$\leq \sigma \left( \frac{1}{4K\gamma} \right) \left( \lambda f \left( \mathcal{G}_{k+1}, \left\{ \mathbf{A}_{k+1}^{(m)} \right\}_1^{N-1}, \tilde{\mathbf{A}}^{(N)} \left( \frac{1}{4K\gamma} \right) \right) - f \left( \mathcal{G}_k, \left\{ \mathbf{A}_k^{(m)} \right\}_1^N \right) + f \left( \mathcal{G}_{k+1}, \left\{ \mathbf{A}_{k+1}^{(m)} \right\}_1^{N-1}, \mathbf{A}_k^{(N)} \right) \right)$

with $\sigma(\eta) = \eta - \frac{1}{4K^\gamma}$ by definition (see the equation (3)).
Since $\sigma\left(\frac{1}{4K^\gamma}\right) = 0$ by definition, the previous inequality yields:

$$\sigma\left(\eta_k^N\right)\left(\lambda f\left(\boldsymbol{\mathcal{G}}_{k+1}, \left\{\mathbf{A}_{k+1}^{(m)}\right\}_1^{N-1}, \tilde{\mathbf{A}}^{(N)}(\eta_k^N)\right) - f\left(\boldsymbol{\mathcal{G}}_k, \left\{\mathbf{A}_k^{(m)}\right\}_1^N\right) + f\left(\boldsymbol{\mathcal{G}}_{k+1}, \left\{\mathbf{A}_{k+1}^{(m)}\right\}_1^{N-1}, \mathbf{A}_k^{(N)}\right)\right) \leq 0$$

Since $\eta_k^N > \frac{1}{4K^\gamma}$ by **Assumption 3.1**, $\sigma(\eta_k^N) = \eta_k^N - \frac{1}{4K^\gamma} > 0$. Thus, the previous inequality yields:

$$\lambda f\left(\boldsymbol{\mathcal{G}}_{k+1}, \left\{\mathbf{A}_{k+1}^{(m)}\right\}_1^{N-1}, \tilde{\mathbf{A}}^{(N)}(\eta_k^N)\right) - f\left(\boldsymbol{\mathcal{G}}_k, \left\{\mathbf{A}_k^{(m)}\right\}_1^N\right) + f\left(\boldsymbol{\mathcal{G}}_{k+1}, \left\{\mathbf{A}_{k+1}^{(m)}\right\}_1^{N-1}, \mathbf{A}_k^{(N)}\right) \leq 0$$

$$\Rightarrow \lambda f\left(\boldsymbol{\mathcal{G}}_{k+1}, \left\{\mathbf{A}_{k+1}^{(m)}\right\}_1^N\right) - f\left(\boldsymbol{\mathcal{G}}_k, \left\{\mathbf{A}_k^{(m)}\right\}_1^N\right) + f\left(\boldsymbol{\mathcal{G}}_{k+1}, \left\{\mathbf{A}_{k+1}^{(m)}\right\}_1^{N-1}, \mathbf{A}_k^{(N)}\right) \leq 0 \ (*)$$

The last implication (*) stems from the definition of $\eta_k^N$ because:
$\tilde{\mathbf{A}}^{(N)}(\eta_k^N) = \mathbf{A}_k^{(N)} - \frac{\eta_k^N}{B_k}\sum_{j\in\mathcal{SET}_k}\partial_{\mathbf{A}^{(N)}}\ell_j = \mathbf{A}_{k+1}^{(N)}$ with $\ell_j$ being defined by (3)) $\qquad\square$

The next result states that $f\left(\boldsymbol{\mathcal{G}}_k, \left\{\mathbf{A}_k^{(m)}\right\}_1^N\right)$ converges to $f_{\min}$ when $k \to \infty$.

The main idea is to prove that $\Delta_k = f\left(\boldsymbol{\mathcal{G}}_k, \left\{\mathbf{A}^{(m)}\right\}_k\right) - f_{min}$ is a recursive sequence verifying the inequality
$(1+\lambda)\Delta_{k+1} \leq \Delta_k + \frac{\zeta(\rho,\alpha,I_n)}{K^\gamma}$ and the conclusion comes from a straightforward reasoning by induction.

---

**Theorem 1** (Convergence to the set of minimizers for *Singleshotinexact* at the rate $\mathcal{O}\left(\frac{1}{k}\right)$). *] The following inequality holds:*

$$\forall k > k_0, f\left(\boldsymbol{\mathcal{G}}_k, \left\{\mathbf{A}_k^{(m)}\right\}_1^N\right) - f_{\min} \leq \frac{f\left(\boldsymbol{\mathcal{G}}_1, \left\{\mathbf{A}_1^{(m)}\right\}_1^N\right) - f_{\min} + \frac{4I_n^2(\rho+\alpha^{N+1})^2\alpha^{2N}}{\lambda}}{k - k_0}$$

*with:*

$$k_0 = 1 + \frac{1}{\log(1+\lambda)}\log\left(\frac{1}{\log(1+\lambda)}\right), \lambda > 0$$

---

*Proof.* Let's denote $f_N$ the function defined by $\mathbf{A}^{(N)} \to f\left(\boldsymbol{\mathcal{G}}_{k+1}, \left\{\mathbf{A}_{k+1}^{(m)}\right\}_1^{N-1}, \mathbf{A}^{(N)}\right) - f_{\min}$
and $\partial f_N\left(\mathbf{A}^{(N)}\right)$ the derivative of $f_N$. Since $\mathbf{A}^{(N)} \to \partial f_N(\mathbf{A}^{(N)})$ is Lipschitz with $\alpha^{2N}$ as the Lipschitz parameter (**Assumption 2** and **Property** 8), we have [10]:

$$f_N\left(\mathbf{A}_{k+1}^{(N)}\right) \leq f_N\left(\mathbf{A}_k^{(N)}\right) + \langle\partial f_N\left(\mathbf{A}_k^{(N)}\right), \mathbf{A}_{k+1}^{(N)} - \mathbf{A}_k^{(N)}\rangle + \frac{\alpha^{2N}}{2}\|\mathbf{A}_{k+1}^{(N)} - \mathbf{A}_k^{(N)}\|_F^2 \quad (6)$$

By definition, we have:
$\mathbf{A}_{k+1}^{(N)} - \mathbf{A}_k^{(N)} = -\eta_k^N\left(\partial f_N(\mathbf{A}_k^{(N)}) + \beta_k\right)$ with $\beta_k = \frac{1}{B_k}\sum_{j\in\mathcal{SET}_k}\partial_{\mathbf{A}^{(N)}}\ell_j - \partial f_N(\mathbf{A}_k^{(N)})$,
$\partial_{\mathbf{A}^{(N)}}\ell_j$ being the derivative of the function $\ell_j$ evaluated at $\mathbf{A}_k^{(N)}$ with:
$\ell_j(\mathbf{A}^{(N)}) = \frac{1}{2}\|\boldsymbol{\mathcal{X}}_j^n - \boldsymbol{\mathcal{G}}_{k+1} \underset{p\in\mathbf{I}_{n-1}}{\times_p} \mathbf{A}_{k+1}^{(p)} \times_n \left(\mathbf{A}_{k+1}^{(n)}\right)_{j,:} \underset{q\in\mathbf{I}_{N-1}^{n+1}}{\times_q} \mathbf{A}_{k+1}^{(q)} \times_N \mathbf{A}^{(N)}\|_F^2$
This equality combined with the inequality (6) yields (since $\eta_k^N > 0$):

$$\frac{f_N\left(\mathbf{A}_{k+1}^{(N)}\right) - f_N\left(\mathbf{A}_k^{(N)}\right)}{\eta_k^N} \leq -\|\partial f_N\left(\mathbf{A}_k^{(N)}\right)\|_F^2 - \langle\partial f_N\left(\mathbf{A}_k^{(N)}\right), \beta_k\rangle + \frac{\alpha^{2N}\eta_k^N}{2}\|\partial f_N\left(\mathbf{A}_k^{(N)}\right) + \beta_k\|_F^2$$

By developing the terms $\|\partial f_N\left(\mathbf{A}_k^{(N)}\right) + \beta_k\|_F^2$ and rearranging them in the last inequality, we have:

$$\frac{f_N\left(\mathbf{A}_{k+1}^{(N)}\right) - f_N\left(\mathbf{A}_k^{(N)}\right)}{\eta_k^N} \leq \frac{\alpha^{2N}\eta_k^N - 2}{2}\|\partial f_N\left(\mathbf{A}_k^{(N)}\right)\|_F^2 + \frac{\alpha^{2N}\eta_k^N}{2}\|\beta_k\|_F^2 + (\alpha^{2N}\eta_k^N - 1)\langle\partial f_N\left(\mathbf{A}_k^{(N)}\right), \beta_k\rangle$$

(7)

Since $\alpha^{2N}\eta_k^N - 1 \leq 0$ (by **Assumption 3.1**) and given that the absolute value of $a$ is equal to $-a$ if $a < 0$, we have by the Cauchy-Schwartz inequality:

$(\alpha^{2N}\eta_k^N - 1)\langle\partial f_N\left(\mathbf{A}_k^{(N)}\right), \beta_k\rangle \leq (1 - \alpha^{2N}\eta_k^N)\|\partial f_N\left(\mathbf{A}_k^{(N)}\right)\|_F\|\beta_k\|_F$

Since $2ab \leq a^2 + b^2$, the previous inequality yields:

$$(\alpha^{2N}\eta_k^N - 1)\langle\partial f_N\left(\mathbf{A}_k^{(N)}\right), \beta_k\rangle \leq \frac{1 - \alpha^{2N}\eta_k^N}{2}(\|\partial f_N\left(\mathbf{A}_k^{(N)}\right)\|_F^2 + \|\beta_k\|_F^2)$$

(8)

The combination of the inequalities (7) and (8) yields after simplification:

$$\frac{f_N\left(\mathbf{A}_{k+1}^{(N)}\right) - f_N\left(\mathbf{A}_k^{(N)}\right)}{\eta_k^N} \leq -\frac{1}{2}\|\partial f_N\left(\mathbf{A}_k^{(N)}\right)\| + \frac{1}{2}\|\beta_k\|_F^2 \leq \frac{1}{2}\|\beta_k\|_F^2$$

Thus, we have:

$$f_N\left(\mathbf{A}_{k+1}^{(N)}\right) \leq f_N\left(\mathbf{A}_k^{(N)}\right) + \frac{\eta_k^N}{2}\|\beta_k\|_F^2$$

Let's consider the sequence $\Delta_k$ defined by: $\Delta_k = f\left(\boldsymbol{\mathcal{G}}_k, \left\{\mathbf{A}_k^{(m)}\right\}_1^N\right) - f_{\min}$. By the replacement of $f_N$ by its expression, the last inequality yields:

$$\Delta_{k+1} \leq \Delta_k + \left(f\left(\boldsymbol{\mathcal{G}}_{k+1}, \left\{\mathbf{A}_{k+1}^{(m)}\right\}_1^{N-1}, \mathbf{A}_k^{(N)}\right) - f\left(\boldsymbol{\mathcal{G}}_k, \left\{\mathbf{A}_k^{(m)}\right\}_1^N\right)\right) + \frac{\eta_k^N}{2}\|\beta_k\|_F^2 \quad (9)$$

By **Property** 13, we have:

$$f\left(\boldsymbol{\mathcal{G}}_{k+1}, \left\{\mathbf{A}_{k+1}^{(m)}\right\}_1^{N-1}, \mathbf{A}_k^{(N)}\right) - f\left(\boldsymbol{\mathcal{G}}_k, \left\{\mathbf{A}_k^{(m)}\right\}_1^N\right) \leq -\lambda f\left(\boldsymbol{\mathcal{G}}_{k+1}, \left\{\mathbf{A}_{k+1}^{(p)}\right\}_1^N\right)$$

$$\leq \underbrace{-\lambda f\left(\boldsymbol{\mathcal{G}}_{k+1}, \left\{\mathbf{A}_{k+1}^{(p)}\right\}_1^N\right) + \lambda f_{\min}}_{-\lambda\Delta_{k+1}}$$

By combining the last inequality with inequality (9), we have

$$(1 + \lambda)\Delta_{k+1} \leq \Delta_k + \frac{\eta_k^N}{2}\|\beta_k\|_F^2$$

By **Property** 12, the last inequality yields:

$$(1 + \lambda)\Delta_{k+1} \leq \Delta_k + \frac{\eta_k^N}{2} \times (8I_n^2(\rho + \alpha^{N+1})^2\alpha^{2N})$$

Given that $\eta_k^N \leq \frac{1}{K^\gamma}$, the last inequality implies:

$$(1 + \lambda)\Delta_{k+1} \leq \Delta_k + \frac{1}{2K^\gamma} \times (8I_n^2(\rho + \alpha^{N+1})^2\alpha^{2N})$$

By introducing, $\tilde{\epsilon}(\rho, \alpha, I_n) = 8I_n^2(\rho + \alpha^{N+1})^2\alpha^{2N}$, this recursive expression yields (by a reasoning by induction):

$\Delta_k \leq \frac{\Delta_1}{(1+\lambda)^{k-1}} + \frac{\tilde{\epsilon}(\rho,\alpha,I_n)}{2K^\gamma}\sum_{m=1}^{k-1}\frac{1}{(1+\lambda)^m} = \frac{\Delta_1}{(1+\lambda)^{k-1}} + \frac{\tilde{\epsilon}(\rho,\alpha,I_n)}{2K^\gamma}\frac{1}{\lambda}\left(1 - \frac{1}{(1+\lambda)^{k-1}}\right)$

$\Rightarrow \Delta_k \leq \frac{\Delta_1}{(1+\lambda)^{k-1}} + \frac{\tilde{\epsilon}(\rho,\alpha,I_n)}{2\lambda K^\gamma}$

Given that $k \leq K$, we have:

$$f\left(\mathcal{G}_k, \left\{\mathbf{A}_k^{(m)}\right\}_1^N\right) - f_{\min} \leq \frac{f\left(\mathcal{G}_1, \left\{\mathbf{A}_1^{(m)}\right\}_1^N\right) - f_{\min}}{(1+\lambda)^{k-1}} + \frac{\tilde{\epsilon}(\rho,\alpha,I_n)}{2\lambda} \times \frac{1}{k^\gamma}$$

Since $\gamma > 1$ by definition, we have:

$$f\left(\mathcal{G}_k, \left\{\mathbf{A}_k^{(m)}\right\}_1^N\right) - f_{\min} \leq \frac{f\left(\mathcal{G}_1, \left\{\mathbf{A}_1^{(m)}\right\}_1^N\right) - f_{\min}}{(1+\lambda)^{k-1}} + \frac{\tilde{\epsilon}(\rho,\alpha,I_n)}{2\lambda} \times \frac{1}{k} \qquad (10)$$

By considering that $k > k_0 = 1 + \frac{1}{log(1+\lambda)}log(\frac{1}{log(1+\lambda)})$ (which is natural since we study the asymptotic behavior with respect to $k$), the equation (10) yields by **Property 10**:

$$f\left(\mathcal{G}_k, \left\{\mathbf{A}_k^{(m)}\right\}_1^N\right) - f_{\min} \leq \frac{f\left(\mathcal{G}_1, \left\{\mathbf{A}_1^{(m)}\right\}_1^N\right) - f_{\min}}{k - k_0} + \frac{\tilde{\epsilon}(\rho,\alpha,I_n)}{2\lambda} \times \frac{1}{k - k_0} \qquad (11)$$

$\square$

### 2.3.2 Convergence of *Singleshot*

The objective of this section to establish for *Singleshot* an ergodic convergence rate of $\mathcal{O}\left(\frac{1}{\sqrt{K}}\right)$. Before establishing this rate, we introduce some preliminary results that are **Properties** 14.1 and 14.2, **Property** 15.2, **Property** 15.1, **Property** 16.

The properties 14.1 and 14.2 simply state that the objective function decreases after each update stage provided the descent steps have been carefully chosen via the minimization problems given by the equations (4) and (5)

**Property 14.1** (Sufficient decrease of the objective function ($\mathcal{G}$) ). *Under **Assumption 4**, we have:*

$$f\left(\mathcal{G}_{k+1}, \left\{\mathbf{A}_k^{(m)}\right\}_1^N\right) \leq f\left(\mathcal{G}_k, \left\{\mathbf{A}_k^{(m)}\right\}_1^N\right)$$

*Proof.* Let's note $f_{\mathcal{G}}^k$ the function $f\left(\cdot, \left\{\mathbf{A}_k^{(m)}\right\}_1^N\right) - f\left(\mathcal{G}_k, \left\{\mathbf{A}_k^{(m)}\right\}_1^N\right)$, we have by the definition of $\eta_k^{\mathcal{G}}$ (equation (4)) and the definition of a minimizer:

$$(\eta_k^{\mathcal{G}} - \frac{\delta_1}{K^{\frac{1}{2}}})f_{\mathcal{G}}^k\left(\mathcal{G}_k - \eta_k^{\mathcal{G}}\mathcal{D}_k^{\mathcal{G}}\right) \leq (\frac{\delta_1}{K^{\frac{1}{2}}} - \frac{\delta_1}{K^{\frac{1}{2}}})f_{\mathcal{G}}^k\left(\mathcal{G}_k - \frac{\delta_1}{K^{\frac{1}{2}}}\mathcal{D}_k^{\mathcal{G}}\right) = 0$$

Since $\eta_k^{\mathcal{G}} > \frac{\delta_1}{K^{\frac{1}{2}}}$ by **Assumption 4**, we have

$$f_{\mathcal{G}}^k\left(\mathcal{G}_k - \eta_k^{\mathcal{G}}\mathcal{D}_k^{\mathcal{G}}\right) \leq 0$$

By definition of $f_{\mathcal{G}}^k$ and given that $\mathcal{G}_{k+1} = \mathcal{G}_k - \eta_k^{\mathcal{G}}\mathcal{D}_k^{\mathcal{G}}$, we have:

$$f\left(\mathcal{G}_{k+1}, \left\{\mathbf{A}_k^{(m)}\right\}_1^N\right) \leq f\left(\mathcal{G}_k, \left\{\mathbf{A}_k^{(m)}\right\}_1^N\right)$$

$\square$

**Property 14.2** (Sufficient decrease of the objective function ($\mathbf{A}^{(p)}$)). *Under **Assumption 4**, we have:*

$$f\left(\mathcal{G}_{k+1}, \left\{\mathbf{A}_{k+1}^{(m)}\right\}_1^p, \left\{\mathbf{A}_k^{(q)}\right\}_{p+1}^N\right) \leq f\left(\mathcal{G}_{k+1}, \left\{\mathbf{A}_{k+1}^{(m)}\right\}_1^{p-1}, \left\{\mathbf{A}_k^{(q)}\right\}_p^N\right)$$

*Proof.* This is the same reasoning as Property 14.1. $\square$

The next two properties simply use the definition of the update schemes

**Property 15.1** (Implicit assumption on $\mathbf{A}_{k+1}^{(p)}$). *Under **Assumption 4**, $\mathbf{A}_{k+1}^{(p)}$ belongs to a ball centered around $\mathbf{A}_k^{(p)}$ whose radius is equal to $\frac{\beta}{K^{\frac{1}{2}}}$.*

*Proof.* By definition, we have:

$$\|\mathbf{A}_{k+1}^{(p)} - \mathbf{A}_k^{(p)}\|_F = \|\tfrac{\delta_2}{K^{\frac{1}{2}}} \partial_{\mathbf{A}^{(p)}} f \left( \mathcal{G}_{k+1}, \left\{ \mathbf{A}_{k+1}^{(m)} \right\}_1^{p-1}, \left\{ \mathbf{A}_k^{(q)} \right\}_p^N \right)\|_F$$

$$\Rightarrow \|\mathbf{A}_{k+1}^{(p)} - \mathbf{A}_k^{(p)}\|_F \leq \tfrac{\delta_2}{K^{\frac{1}{2}}} \Gamma_p \text{ (by definition of } \Gamma_p)$$

$$\Rightarrow \|\mathbf{A}_{k+1}^{(p)} - \mathbf{A}_k^{(p)}\|_F \leq \tfrac{\delta_2}{K^{\frac{1}{2}}} \max(\Gamma_g, \Gamma_1, ..., \Gamma_N) \qquad \qquad \square$$

**Property 15.2** (Implicit assumption on $\mathcal{G}_{k+1}$). *Under **Assumption 4**, $\mathcal{G}_{k+1}$ belongs to a ball centered around $\mathcal{G}_k$ whose radius is equal to $\frac{\beta}{K^{\frac{1}{2}}}$.*

*Proof.* With the same reasoning as for Property 15.1, we have:

$$\|\mathcal{G}_{k+1} - \mathcal{G}_k\|_F \leq \frac{\delta_2}{K^{\frac{1}{2}}} \Gamma_g \leq \frac{\delta_2}{K^{\frac{1}{2}}} \max(\Gamma_g, \Gamma_1, ..., \Gamma_N)$$

$$\square$$

The next property establishes the Lipschitz property for the squared Frobenius of the block-wise derivative (which will be needed in **Property** 17.1)

**Property 16.** *Lipschitz property for the squared Frobenius of the block-wise derivative Let's consider the function $g_p$ defined by:*

$$g_p \left( \mathcal{G}, \mathbf{A}^{(1)}, .., \mathbf{A}^{(N)} \right) = \|\partial_{\mathbf{A}^{(p)}} f \left( \mathcal{G}, \mathbf{A}^{(1)}, .., \mathbf{A}^{(N)} \right)\|_F^2, 1 \leq p \leq N \qquad (12)$$

*We consider $g_p$ is defined on the set $\widehat{\mathbb{D}}_g \times \widehat{\mathbb{D}}_1 \times ... \times \widehat{\mathbb{D}}_N$ with $\widehat{\mathbb{D}}_g = \left\{ \mathcal{G} \in \mathbb{R}^{J_1 \times ... \times J_N} | \|\mathcal{G}\|_F < 2\alpha \right\}, \widehat{\mathbb{D}}_m = \left\{ \mathbf{A}^{(m)} \in \mathbb{R}^{I_m \times J_m} | \|\mathbf{A}^{(m)}\|_F < 2\alpha \right\}$. The function $f_p$ is Lipschitz (therefore, the restriction of $g_p$ to the set $\mathbb{D}_g \times \mathbb{D}_1 \times .... \times \mathbb{D}_N$ is Lipschitz since $\mathbb{D}_g \times \mathbb{D}_1 \times .... \times \mathbb{D}_N \subset \widehat{\mathbb{D}}_g \times \widehat{\mathbb{D}}_1 \times ... \times \widehat{\mathbb{D}}_N$ )*

*Proof.* by noticing that the derivative of $g_p$ is bounded on $\widehat{\mathbb{D}}_g \times \widehat{\mathbb{D}}_1 \times ... \times \widehat{\mathbb{D}}_N$ and given that a finite product of convex sets (respectively open sets) is a convex set (respectively open set), by [11]:**Corollary 2.31** yields the Lipschitz character of the function $g_p$. $\square$

The next three properties bound the mean block-wise derivative for the core and the loadings factors. They mainly use known algebraic arguments and **Properties** 14.1 or 14.2. As the second property includes indicia $k$ and $k+1$, the third property is necessary to bound the mean block-wise derivative of the loading factors at the $k^{th}$ iteration.

**Property 17.1.** *Mean block-wise derivative ($\mathcal{G}$)*

$$\forall K \geq 1, \frac{1}{K} \sum_{k=0}^{K-1} \|\partial_{\mathcal{G}} f \left( \mathcal{G}_k, \left\{ \mathbf{A}_k^{(m)} \right\}_1^N \right)\|_F^2 \leq \frac{\frac{\delta_2}{\delta_1^2} \left( 2\Gamma + \alpha^{2N} \Gamma_g^2 \delta_2^2 \right)}{K^{\frac{1}{2}}} \qquad (13)$$

*Proof.* By **Assumption 2** (factors boundedness) and **Property** 6 (Lipschitz derivative), we have

$$\|\partial_{\mathcal{G}} f\left(\mathcal{G}_1, \left\{\mathbf{A}_k^{(m)}\right\}_1^N\right) - \partial_{\mathcal{G}} f\left(\mathcal{G}_2, \left\{\mathbf{A}_k^{(m)}\right\}_1^N\right)\|_F \leq \|\mathcal{G}_1 - \mathcal{G}_2\|_F \prod_{m \in I_N} \|\mathbf{A}_k^{(m)}\|_F^2 \leq \alpha^{2N}\|\mathcal{G}_1 - \mathcal{G}_2\|_F$$

Thus, $\partial_{\mathcal{G}} f\left(., \left\{\mathbf{A}_k^{(m)}\right\}_1^N\right)$ is Lipschitz, which paves the way to the inequality [10]:

$$f\left(\mathcal{G}_{k+1}, \left\{\mathbf{A}_k^{(m)}\right\}_1^N\right) \leq f\left(\mathcal{G}_k, \left\{\mathbf{A}_k^{(m)}\right\}_1^N\right) + \langle \partial_{\mathcal{G}} f\left(\mathcal{G}_k, \left\{\mathbf{A}_k^{(m)}\right\}_1^N\right), \mathcal{G}_{k+1} - \mathcal{G}_k \rangle + \frac{\alpha^{2N}}{2}\|\mathcal{G}_{k+1} - \mathcal{G}_k\|_F^2$$

Since $\mathcal{G}_{k+1} = \mathcal{G}_k - \eta_k^{\mathcal{G}} \partial_{\mathcal{G}} f\left(\mathcal{G}_k; \left\{\mathbf{A}_k^{(m)}\right\}_1^N\right)$ and by definition of $\Gamma_g$ as the supremum of $\|\partial_{\mathcal{G}} f\left(\mathcal{G}, \left\{\mathbf{A}^{(m)}\right\}_1^N\right)\|_F$ on $\mathbb{D}_g \times \mathbb{D}_1 \times ... \times \mathbb{D}_N$, we have:

$$\eta_k^{\mathcal{G}}\|\partial_{\mathcal{G}} f\left(\mathcal{G}_k, \left\{\mathbf{A}_k^{(m)}\right\}_1^N\right)\|_F^2 \leq f\left(\mathcal{G}_k, \left\{\mathbf{A}_k^{(m)}\right\}_1^N\right) - f\left(\mathcal{G}_{k+1}, \left\{\mathbf{A}_k^{(m)}\right\}_1^N\right) + \frac{\alpha^{2N}(\eta_k^{\mathcal{G}})^2}{2}\Gamma_g^2$$

$$(14)$$

By **Properties** 14.1 and 14.2, we have $f\left(\mathcal{G}_{k+1}, \left\{\mathbf{A}_{k+1}^{(m)}\right\}_1^N\right) \leq f\left(\mathcal{G}_{k+1}, \left\{\mathbf{A}_k^{(m)}\right\}_1^N\right)$. This inequality combined with the inequality (14) yields

$$\eta_k^{\mathcal{G}}\|\partial_{\mathcal{G}} f\left(\mathcal{G}_k, \left\{\mathbf{A}_k^{(m)}\right\}_1^N\right)\|_F^2 \leq f\left(\mathcal{G}_k, \left\{\mathbf{A}_k^{(m)}\right\}_1^N\right) - f\left(\mathcal{G}_{k+1}, \left\{\mathbf{A}_{k+1}^{(m)}\right\}_1^N\right) + \frac{\alpha^{2N}\Gamma_g^2}{2}(\eta_k^{\mathcal{G}})^2$$

By taking the sum of this inequality, we have:

$$\sum_{k=0}^{K-1} \eta_k^{\mathcal{G}}\|\partial_{\mathcal{G}} f\left(\mathcal{G}_k, \left\{\mathbf{A}_k^{(m)}\right\}_1^N\right)\|_F^2 \leq f(\mathcal{G}_0, \left\{\mathbf{A}_0^{(m)}\right\}_1^N) - f\left(\mathcal{G}_K, \left\{\mathbf{A}_K^{(m)}\right\}_1^N\right) + \frac{\alpha^{2N}\Gamma_g^2}{2}\sum_{k=0}^{K-1}(\eta_k^{\mathcal{G}})^2$$

$$\Rightarrow \sum_{k=0}^{K-1} \eta_k^{\mathcal{G}}\|\partial_{\mathcal{G}} f\left(\mathcal{G}_k, \left\{\mathbf{A}_k^{(m)}\right\}_1^N\right)\|_F^2 \leq 2\Gamma + \frac{\alpha^{2N}\Gamma_g^2}{2}\sum_{k=0}^{K-1}(\eta_k^{\mathcal{G}})^2 \text{(by definition of } \Gamma \text{ as the supremum of } f)$$

Since $\frac{\delta_1}{K^{\frac{1}{2}}} < \eta_{g,k} \leq \frac{\delta_2}{K^{\frac{1}{2}}}$, we have

$$\frac{1}{\sum_{k=0}^{K-1} \eta_k^{\mathcal{G}}} \leq \frac{1}{\delta_1 K^{1/2}}, \qquad\qquad \sum_{k=0}^{K-1} \frac{(\eta_k^{\mathcal{G}})^2}{\sum_{k=0}^{K-1} \eta_k^{\mathcal{G}}} \leq \frac{\delta_2^2}{\delta_1 K^{1/2}}$$

and

$$\sum_{k=0}^{K-1} \frac{\eta_k^{\mathcal{G}}}{\sum_{k=0}^{K-1} \eta_k^{\mathcal{G}}}\|\partial_{\mathcal{G}} f\left(\mathcal{G}_k, \left\{\mathbf{A}_k^{(m)}\right\}_1^N\right)\|_F^2 \geq \frac{\delta_1}{\delta_2}\frac{1}{K}\sum_{k=0}^{K-1}\|\partial_{\mathcal{G}} f\left(\mathcal{G}_k, \left\{\mathbf{A}_k^{(m)}\right\}_1^N\right)\|_F^2$$

With the last three inequalities, we have:

$$\frac{1}{K}\sum_{k=0}^{K-1}\|\partial_{\mathcal{G}} f\left(\mathcal{G}_k, \left\{\mathbf{A}_k^{(m)}\right\}_1^N\right)\|_F^2 \leq \frac{\delta_2}{\delta_1^2}\frac{\left(2\Gamma + \frac{\alpha^{2N}\Gamma_g^2\delta_2^2}{2}\right)}{K^{\frac{1}{2}}}$$

$\square$

**Property 17.2** (Mean block-wise derivative ($\mathbf{A}^{(p)}$)).

$$\forall K \geq 1, \frac{1}{K}\sum_{k=0}^{K-1}\|\partial_{\mathbf{A}^{(p)}} f\left(\mathcal{G}_{k+1}, \left\{\mathbf{A}_{k+1}^{(m)}\right\}_1^{p-1}, \mathbf{A}_k^{(p)}, \left\{\mathbf{A}_k^{(q)}\right\}_{p+1}^N\right)\|_F^2 \leq \frac{\frac{\delta_2}{\delta_1^2}\left(2\Gamma + \frac{\alpha^{2N}\Gamma_p^2\delta_2^2}{2}\right)}{K^{\frac{1}{2}}}$$

*Proof.* Let's note $f_k^{(p)}$ the function $f\left(\mathcal{G}_{k+1}, \left\{\mathbf{A}_{k+1}^{(m)}\right\}_1^{p-1}, \cdot, \left\{\mathbf{A}_k^{(q)}\right\}_{p+1}^{N}\right)$. Since the reasoning is identical to the one previously performed before the equation (14), and given that the Lipschitz constant for the derivative with respect to $\mathbf{A}^{(m)}$ is also equal to $\alpha^{2N}$ (by **Property** 8 and **Assumption** 2), we have:

$$\eta_k^p \|\partial_{\mathbf{A}^{(p)}} f\left(\mathcal{G}_{k+1}, \left\{\mathbf{A}_{k+1}^{(m)}\right\}_1^{p-1}, \mathbf{A}_k^{(p)}, \left\{\mathbf{A}_k^{(q)}\right\}_{p+1}^{N}\right)\|_F^2 \leq -f_k^{(p)}\left(\mathbf{A}_{k+1}^{(p)}\right) + f_k^{(p)}\left(\mathbf{A}_k^{(p)}\right) + \frac{\alpha^{2N}\left(\eta_k^p\right)^2 \Gamma_p^2}{2}$$

(15)

By **Properties** 14.1 and 14.2, we have

$$f\left(\mathcal{G}_{k+1}, \left\{\mathbf{A}_{k+1}^{(m)}\right\}_1^N\right) \leq f\left(\mathcal{G}_{k+1}, \left\{\mathbf{A}_{k+1}^{(m)}\right\}_1^{p-1}, \mathbf{A}_{k+1}^{(p)}, \left\{\mathbf{A}_k^{(q)}\right\}_{p+1}^N\right) = f_k^{(p)}\left(\mathbf{A}_{k+1}^{(p)}\right),$$

and

$$f_k^{(p)}\left(\mathbf{A}_k^{(p)}\right) = f\left(\mathcal{G}_{k+1}, \left\{\mathbf{A}_{k+1}^{(m)}\right\}_1^{p-1}, \mathbf{A}_k^{(p)}, \left\{\mathbf{A}_k^{(q)}\right\}_{p+1}^N\right) \leq f\left(\mathcal{G}_k, \left\{\mathbf{A}_k^{(m)}\right\}_1^N\right).$$

By inserting the last two inequalities in the equation (15), we have:

$$\eta_k^p \|\partial_{\mathbf{A}^{(p)}} f\left(\mathcal{G}_{k+1}, \left\{\mathbf{A}_{k+1}^{(m)}\right\}_1^{p-1}, \mathbf{A}_k^{(p)}, \left\{\mathbf{A}_k^{(q)}\right\}_{p+1}^N\right)\|_F^2 \qquad \leq \qquad f\left(\mathcal{G}_k, \left\{\mathbf{A}_k^{(m)}\right\}_1^N\right) \quad -$$
$$f\left(\mathcal{G}_{k+1}, \left\{\mathbf{A}_{k+1}^{(m)}\right\}_1^N\right) + \frac{\alpha^{2N}\Gamma_p^2}{2}(\eta_k^p)^2$$

From this point, we perform exactly a reasoning identical to the one performed for **Property** 17.1 (from inequality (14)) and get the result. □

The idea of the proof of the following theorem is similar to the classical reasoning used to prove the "sequential criterion for continuity".

**Property 17.3** (Bound on mean derivative for any mode).

$$\exists K_p \geq 1, \forall K \geq K_p, \frac{1}{K}\sum_{k=0}^{K-1}\|\partial_{\mathbf{A}^{(p)}} f\left(\mathcal{G}_k, \left\{\mathbf{A}_k^{(m)}\right\}_1^N\right)\|_F^2 \leq \frac{\delta_2}{\delta_1^2 K^{\frac{1}{2}}}\left(1 + \left(2\Gamma + \frac{\alpha^{2N}\Gamma_p^2\delta_2^2}{2}\right)\right)$$

(16)

*Proof.* The function $g_p$ defined by $g_p\left(\mathcal{G}, \mathbf{A}^{(1)}, .., \mathbf{A}^{(N)}\right) = \|\partial_{\mathbf{A}^{(p)}} f\left(\mathcal{G}, \mathbf{A}^{(1)}, \cdots, \mathbf{A}^{(N)}\right)\|_F^2$ being lipschitz by the equation **Property** 16, it is uniformly continuous. Then, we have:

$$\forall \epsilon > 0, \exists \eta > 0, \|x - y\|_* \leq \eta \Rightarrow |\|\partial_{\mathbf{A}^{(p)}} f(x)\|_F^2 - \|\partial_{\mathbf{A}^{(p)}} f(y)\|_F^2| \leq \epsilon \qquad (17)$$

Let's consider $y = (\mathcal{G}_{k+1}, \mathbf{A}_{k+1}^{(1)}, \cdots, \mathbf{A}_{k+1}^{(p-1)}, \mathbf{A}_k^{(p)}, \cdots, \mathbf{A}_k^{(N)})$ and $x = \left(\mathcal{G}_k, \mathbf{A}_k^{(1)}, \cdots, \mathbf{A}_k^{(N)}\right)$

$\|x - y\|_* = \|\mathcal{G}_{k+1} - \mathcal{G}_k\|_F + \|\mathbf{A}_{k+1}^{(1)} - \mathbf{A}_k^{(1)}\|_F + .... + \|\mathbf{A}_{k+1}^{(p-1)} - \mathbf{A}_k^{(p-1)}\|_F \leq \frac{p\beta}{K^{\frac{1}{2}}}$ (by **Property** 15.2 and 15.1 )

Thus, we have:

$$\|x - y\|_* \leq \frac{p\beta}{K^{\frac{1}{2}}} \qquad (18)$$

Since $\frac{p\beta}{K^{\frac{1}{2}}}$ converges to zero when $K \to \infty$, $\exists K_p, \forall K \geq K_p, \|x-y\|_* \leq \frac{p\beta}{K^{\frac{1}{2}}} \leq \eta$. By the inequality (17), we have:

$\forall \epsilon > 0, \exists K_p, \forall K \geq K_p, |\|\partial_{\mathbf{A}^{(p)}} f(x)\|_F^2 - \|\partial_{\mathbf{A}^{(p)}} f(y)\|_F^2| \leq \epsilon$
$\Rightarrow \forall \epsilon > 0, \exists K_p, \forall K \geq K_p, \|\partial_{\mathbf{A}^{(p)}} f(x)\|_F^2 - \|\partial_{\mathbf{A}^{(p)}} f(y)\|_F^2 \leq \epsilon$ (because $a \leq |a|$)
$\Rightarrow \forall \epsilon > 0, \exists K_p, \forall K \geq K_p, \|\partial_{\mathbf{A}^{(p)}} f(x)\|_F^2 \leq \epsilon + \|\partial_{\mathbf{A}^{(p)}} f(y)\|_F^2$
$\Rightarrow \forall \epsilon > 0, \exists K_p, \forall K \geq K_p, \|\partial_{\mathbf{A}^{(p)}} f\left(\mathcal{G}_k, \left\{\mathbf{A}_k^{(m)}\right\}_1^N\right)\|_F^2 \leq \epsilon$

$$+\|\partial_{\mathbf{A}^{(p)}} f\left(\mathcal{G}_{k+1}, \left\{\mathbf{A}_{k+1}^{(m)}\right\}_1^{p-1}, \left\{\mathbf{A}_k^{(q)}\right\}_p^N\right)\|_F^2$$

The last inequality holds for any value of $k \in \{1, 2, ..K\}$ provided $K \geq K_p$ (it is worth to notice that we are dealing with two variables $k$ and $K$ that are different). Then, by summing over $k$ (not $K$), we have: $\forall \epsilon > 0, \exists K_p, \forall K \geq K_p$:

$$\frac{1}{K}\sum_{k=0}^{K-1}\|\partial_{\mathbf{A}^{(p)}} f\left(\boldsymbol{\mathcal{G}}_k, \left\{\mathbf{A}_k^{(p)}\right\}_1^N\right)\|_F^2 \leq \epsilon + \frac{1}{K}\sum_{k=0}^{K-1}\|\partial_{\mathbf{A}^{(p)}} f\left(\boldsymbol{\mathcal{G}}_{k+1}, \left\{\mathbf{A}_{k+1}^{(m)}\right\}_1^{p-1}, \left\{\mathbf{A}_k^{(q)}\right\}_p^N\right)\|_F^2$$

$$\Rightarrow \forall \epsilon > 0, \exists K_p, \forall K \geq K_p, \frac{1}{K}\sum_{k=0}^{K-1}\|\partial_{\mathbf{A}^{(p)}} f(\boldsymbol{\mathcal{G}}_k, \left\{\mathbf{A}_k^{(m)}\right\}_1^N)\|_F^2 \leq \epsilon + \frac{\frac{\delta_2}{\delta_1^2}(2\Gamma + \frac{\alpha^{2N}\Gamma_p^2\delta_2^2}{2})}{K^{\frac{1}{2}}} \text{ (by Property 17.2)}$$

$$\Rightarrow \exists K_p \geq 1, \forall K \geq K_p, \frac{1}{K}\sum_{k=0}^{K-1}\|\partial_{\mathbf{A}^{(p)}} f\left(\boldsymbol{\mathcal{G}}_k, \left\{\mathbf{A}_k^{(m)}\right\}_1^N\right)\|_F^2 \leq \frac{\frac{\delta_2}{\delta_1^2}}{K^{\frac{1}{2}}} + \frac{\frac{\delta_2}{\delta_1^2}(2\Gamma + \frac{\alpha^{2N}\Gamma_p^2\delta_2^2}{2})}{K^{\frac{1}{2}}}$$

Thus, we have:

$$\exists K_p \geq 1, \forall K \geq K_p, \frac{1}{K}\sum_{k=0}^{K-1}\|\partial_{\mathbf{A}^{(p)}} f\left(\boldsymbol{\mathcal{G}}_k, \left\{\mathbf{A}_k^{(m)}\right\}_1^N\right)\|_F^2 \leq \frac{\delta_2}{\delta_1^2 K^{\frac{1}{2}}}\left(1 + \left(2\Gamma + \frac{\alpha^{2N}\Gamma_p^2\delta_2^2}{2}\right)\right) \tag{19}$$

$\square$

---

**Theorem 2.** *Convergence rate*
*The following inequality holds:*

$$\exists K_0 > 0, \forall K \geq K_0, \frac{1}{K}\sum_{k=0}^{K-1}\|\nabla f\left(\boldsymbol{\mathcal{G}}_k, \left\{\mathbf{A}_k^{(m)}\right\}_1^N\right)\|_*^2 \leq \frac{(N+1)\Delta}{K^{\frac{1}{2}}} \tag{20}$$

*with* $\Delta = \frac{\delta_2}{\delta_1^2}\left(2\Gamma + \frac{\alpha^{2N}\Gamma_g^2\delta_2^2}{2} + \sum_{p=1}^N (1 + 2\Gamma + \frac{\alpha^{2N}\Gamma_p^2\delta_2^2}{2})\right)$

---

*Proof.* First, let's establish a bound on $\frac{1}{K}\sum_{k=0}^{K-1}\|\nabla f\left(\boldsymbol{\mathcal{G}}_k, \left\{\mathbf{A}_k^{(m)}\right\}_1^N\right)\|_*^2$ in order to make appear

$\frac{1}{K}\sum_{k=0}^{K-1}\|\partial_{\boldsymbol{\mathcal{G}}} f\left(\boldsymbol{\mathcal{G}}_k, \left\{\mathbf{A}_k^{(m)}\right\}_1^N\right)\|_F^2, \frac{1}{K}\sum_{k=1}^{K}\|\partial_{\mathbf{A}^{(1)}} f(\boldsymbol{\mathcal{G}}_k, \left\{\mathbf{A}_k^{(m)}\right\}_1^N)\|_F^2, \cdots, \frac{1}{K}\sum_{k=1}^{K}\|\partial_{\mathbf{A}^{(N)}} f(\boldsymbol{\mathcal{G}}_k, \left\{\mathbf{A}_k^{(m)}\right\}_1^N)\|_F^2$, which are the variables that have already been bounded via the inequality and **Properties** 17.1 and 17.3.

By convexity of the function $g(x) = x^2$, since $\|\nabla f\left(\boldsymbol{\mathcal{G}}_k, \left\{\mathbf{A}_k^{(m)}\right\}_1^N\right)\|_*$ is equal to the sum of the norms of the block-wise derivatives evaluated at $\boldsymbol{\mathcal{G}}_k, \mathbf{A}_k^{(1)}, \cdots, \mathbf{A}_k^{(N)}$, we have:

$$\|\nabla f\left(\boldsymbol{\mathcal{G}}_k, \left\{\mathbf{A}_k^{(m)}\right\}_1^N\right)\|_*^2 \leq (N+1)\left(\|\partial_{\boldsymbol{\mathcal{G}}} f(\boldsymbol{\mathcal{G}}_k, \left\{\mathbf{A}_k^{(m)}\right\}_1^N)\|_F^2 + \sum_{p=1}^N \|\partial_{\mathbf{A}^{(p)}} f\left(\boldsymbol{\mathcal{G}}_k, \left\{\mathbf{A}_k^{(m)}\right\}_1^N\right)\|_F^2\right)$$

$$\Rightarrow \frac{1}{K}\sum_{k=0}^{K-1}\|\nabla f\left(\boldsymbol{\mathcal{G}}_k, \left\{\mathbf{A}_k^{(m)}\right\}_1^N\right)\|_*^2 \leq (N+1)\frac{1}{K}\sum_{k=0}^{K-1}\|\partial_{\boldsymbol{\mathcal{G}}} f\left(\boldsymbol{\mathcal{G}}_k, \left\{\mathbf{A}_k^{(m)}\right\}_1^N\right)\|_F^2$$

$$+ (N+1)\sum_{k=0}^{K-1}\frac{1}{K}\sum_{p=1}^N \|\partial_{\mathbf{A}^{(p)}} f\left(\boldsymbol{\mathcal{G}}_k, \left\{\mathbf{A}_k^{(m)}\right\}_1^N\right)\|_F^2$$

By permuting the two signs $\sum$, we have:

$$\Rightarrow \frac{1}{K}\sum_{k=0}^{K-1}\|\nabla f\left(\boldsymbol{\mathcal{G}}_k, \left\{\mathbf{A}_k^{(m)}\right\}_1^N\right)\|_*^2 \leq (N+1)\left(\underbrace{\frac{1}{K}\sum_{k=0}^{K-1}\|\partial_{\boldsymbol{\mathcal{G}}} f(\boldsymbol{\mathcal{G}}_k, \left\{\mathbf{A}_k^{(m)}\right\}_1^N)\|_F^2}_{\text{can be bounded by Property 17.1}}\right) \tag{21}$$

$$+(N+1)\sum_{p=1}^{N}\underbrace{\sum_{k=0}^{K-1}\frac{1}{K}\|\partial_{\mathbf{A}^{(p)}}f\left(\boldsymbol{\mathcal{G}}_k,\left\{\mathbf{A}_k^{(m)}\right\}_1^N\right)\|_F^2}_{\text{can be bounded by \textbf{Property} 17.3}}$$

By **Property** 17.3, we have for each mode $p$:

$$\exists K_p \geq 1, \forall K \geq K_p, \frac{1}{K}\sum_{k=0}^{K-1}\|\partial_{\mathbf{A}^{(p)}}f\left(\boldsymbol{\mathcal{G}}_k,\left\{\mathbf{A}_k^{(m)}\right\}_1^N\right)\|_F^2 \leq \frac{\delta_2}{\delta_1^2 K^{\frac{1}{2}}}\left(1+\left(2\Gamma+\frac{\alpha^{2N}\Gamma_p^2\delta_2^2}{2}\right)\right)$$
(22)

For $K \geq max(K_p, 1 \leq p \leq N)$, all of the inequalities given by (22) are verified for any mode $p$. Thus, the inequality (21) yields:

$$\frac{1}{K}\sum_{k=0}^{K-1}\|\nabla f\left(\boldsymbol{\mathcal{G}}_k,\left\{\mathbf{A}_k^{(m)}\right\}_1^N\right)\|_*^2 \leq (N+1)\left(\frac{1}{K}\sum_{k=0}^{K-1}\|\partial_{\boldsymbol{\mathcal{G}}}f\left(\boldsymbol{\mathcal{G}}_k,\left\{\mathbf{A}_k^{(m)}\right\}_1^N\right)\|_F^2\right)$$

$$+(N+1)\left(\sum_{p=1}^{N}\frac{\delta_2}{\delta_1^2 K^{\frac{1}{2}}}\left(1+\left(2\Gamma+\frac{\alpha^{2N}\Gamma_p^2\delta_2^2}{2}\right)\right)\right)$$

$$\Rightarrow \frac{1}{K}\sum_{k=0}^{K-1}\|\nabla f\left(\boldsymbol{\mathcal{G}}_k,\left\{\mathbf{A}_k^{(m)}\right\}_1^N\right)\|_*^2 \leq (N+1)\left(\frac{\frac{\delta_2}{\delta_1^2}(2\Gamma+\frac{\alpha^{2N}\Gamma_g^2\delta_2^2}{2})}{K^{\frac{1}{2}}}+\sum_{p=1}^{N}\frac{\delta_2}{\delta_1^2 K^{\frac{1}{2}}}\left(1+2\Gamma+\frac{\alpha^{2N}\Gamma_p^2\delta_2^2}{2}\right)\right)$$

The last implication results from **Property** 17.1. Finally, we have:

$$\forall K \geq \max(K_p, 1 \leq p \leq N), \frac{1}{K}\sum_{k=0}^{K-1}\|\nabla f\left(\boldsymbol{\mathcal{G}}_k,\left\{\mathbf{A}_k^{(m)}\right\}_1^N\right)\|_*^2 \leq \frac{(N+1)\Delta}{K^{\frac{1}{2}}}$$
(23)

$\square$

# 3 More details on variants

## 3.1 Singleshot-online

We consider now the problem of decomposing a tensor that can grow over time with respect to every mode with the single pass constraint (i.e. with no need to resort to the past data), hence the term "online". This is a relevant problem that has received little attention [2]. Let's consider a $N-$order tensor growing over time with respect to any of its modes. Let's assume that at time step $t$, its state is $\mathcal{X}_t \in \mathbb{R}^{I_{1,t} \times \dots \times I_{N,t}}$ and that at the time step $t+1$, we have acquired $B_n$ new subtensors with respect to the mode $n$ $Set_n = \left\{ \mathcal{X}_q^n \in \mathbb{R}^{I_{1,t} \times \dots \times I_{n-1,t} \times 1 \times I_{n+1,t} \times \dots \times I_{N,t}}, 1 \le q \le B_n \right\}$. According to the Tucker decomposition expression, $\mathbf{A}^{(n)}$ is the single matrix which dimensions change: its number of rows increases from $I_{n,t}$ to $I_{n,t} + B_n$.

Based on the above described algorithms, we can derive a novel variant able to infer the factors that approximate $\mathcal{X}_{t+1} \in \mathbb{R}^{\prod_{k=1}^{n-1} I_{k,t} \times (I_{n,t}+B_n) \times \prod_{k=n+1}^{N} I_{k,t}}$.

We propose an update scheme identical to the one presented for *Singleshot-inexact*, with the core intuition that the online problem can be efficiently tackled via our inexact gradient approach. More formally, the derivatives with respect to the core tensor $\mathcal{G}$ and a loading matrix $\mathbf{A}^{(p)}, p \neq n$, denoted by $\widehat{\mathcal{D}}_k^{\mathcal{G}}$ and $\widehat{\mathbf{D}}_k^p$ are computed as for the *Singleshot-inexact* method by fixing $\mathcal{SET}_k$ to $Set_n$. The derivative $\widehat{\mathbf{D}}_k^n \in \mathbb{R}^{(I_{n,t}+B_n) \times J_n}$ is defined by:

$$\widehat{\mathbf{D}}_k^n = [\ \underbrace{\mathbf{o}, \cdots, \mathbf{o}}_{I_{n,t} \text{zero vectors}}\ , \mathbf{v}_1, .., \mathbf{v}_{B_n}], \tag{24}$$

with $[., ..., .]$ being the row-wise stacking operator, $\mathbf{o} \in \mathbb{R}^{1 \times J_n}$ being the null vector and $\mathbf{v}_j$ the derivative of $\ell_j$ evaluated at $(\mathbf{A}_k^{(n)})_{I_{n,t}+j,:}$, and $\ell_j$ defined by:

$$\ell_j(\mathbf{a}) = \frac{1}{2} \| \mathcal{X}_j^n - \mathcal{G}_{k+1} \underset{m \in \mathbf{I}_{n-1}}{\times_p} \mathbf{A}_{k+1}^{(m)} \times_n \mathbf{a} \underset{q \in \mathbf{I}_N^{n+1}}{\times_q} \mathbf{A}_k^{(q)} \|_F^2$$

with $\mathcal{X}_j^n \in Set_n, \mathbf{a} \in \mathbb{R}^{1 \times J_n}$.
One can notice that these update patterns require a single pass on the data since they do not involve at all $\mathcal{X}_t$. The initial values of $\mathcal{G}$ and $\mathbf{A}^{(p)}, p \neq n$ are fixed to $\mathcal{G}_t$ and $\mathbf{A}_t^{(p)}$ while the initial value of $\mathbf{A}^{(n)}$ is defined via the row-wise stacking of $\mathbf{A}_t^{(n)}$ with $B_n$ random vectors.

## 3.2 Non-negative constraints

To perform a nonnegative tensor factorization, we simply replace for *Singleshot*,*Singleshot-inexact* and *Singleshot-online*, all the update schemes by *Projected Gradient Descent* [14]:

$$\mathcal{G}_{k+1} \leftarrow \max(\mathcal{G}_k - \eta_{g,k} \mathcal{D}_k^{\mathcal{G}}, 0), \eta_{g,k} > 0$$
$$\mathbf{A}_{k+1}^{(p)} \leftarrow \max(\mathbf{A}_k^{(p)} - \eta_{p,k} \mathbf{D}_k^p, 0), \eta_{p,k} > 0$$

with the tensor $\mathcal{D}_k^{\mathcal{G}}$ and the matrix $\mathbf{D}_k^p$ representing the descent direction for *Singleshot*,*Singleshot-inexact* or *Singleshot-online*.

# 4 Numerical experiments: follow up

## 4.1 More details about the data sets used for the experiments in the main paper

- **Enron**: constructed from the emails of **Enron**, this data set represents a three-order tensor $\mathcal{X} \in \mathbb{R}^{M \times M \times 200}$. The three modes respectively represent the sender, the recipient and the words. The entry $\mathcal{X}_{i,j,k}$ is equal to 1 if the $i^{th}$ sender sends a message to the $j^{th}$ recipient containing the $k^{th}$ words. The words considered are the most frequent ones. The problem considered for this data set is a regression problem, the evaluation criterion being the approximation error on a test set (with the same size as the training set, but obviously

different from this one) defined by:

$$AE = \|\boldsymbol{\mathcal{X}}_{test} - \boldsymbol{\mathcal{G}}_s \times_1 \mathbf{A}_{out}^{(1)} \times_2 \mathbf{A}_{out}^2 \times_3 \mathbf{A}_{out}^{(3)}\|_F \tag{25}$$

The tensor $\boldsymbol{\mathcal{X}}_{test}$ represents the test set, the matrices $\left\{\mathbf{A}_{out}^{(m)}, 1 \le m \le 3\right\}$ the latent factors inferred from the decomposition and the tensor $\boldsymbol{\mathcal{G}}_s$ is defined by:

- For the unconstrained decomposition
  $\boldsymbol{\mathcal{G}}_s \leftarrow \arg\min_{\boldsymbol{\mathcal{G}}} \|\boldsymbol{\mathcal{X}}_{test} - \boldsymbol{\mathcal{G}} \times_1 \mathbf{A}_{out}^{(1)} \times_2 \mathbf{A}_{out}^{(2)} \times_3 \mathbf{A}_{out}^{(3)}\|_F^2$
  This problem is solved by the classical gradient descent algorithm.
- For the positive decomposition
  $\boldsymbol{\mathcal{G}}_s \leftarrow \arg\min_{\boldsymbol{\mathcal{G}} \ge 0} \|\boldsymbol{\mathcal{X}}_{test} - \boldsymbol{\mathcal{G}} \times_1 \mathbf{A}_{out}^{(1)} \times_2 \mathbf{A}_{out}^{(2)} \times_3 \mathbf{A}_{out}^{(3)}\|_F^2$
  This problem is solved by the classical project gradient descent algorithm.

- **Movielens**: is a tensor $\boldsymbol{\mathcal{X}} \in \mathbb{R}^{M \times M \times 610}$ constructed from the MovieLens latest-small data set [4] and whose modes represent the time steps, the movies and the users. The entry $\boldsymbol{\mathcal{X}}_{i,j,k}$ corresponds to the rate given by the $k^{th}$ user to the $j^{th}$ movie at the time step $i$ or zero if no rate has been given. The purpose of this data set is to validate our model on a tensor-based recommender system. The evaluation criterion is the Mean Average Precision *MAP* defined as in [5](section 4.1.2) on the test set.

## 4.2 Singleshot/Singleshotinexact vs Tensorsketchonline on *Enron* and *Movielens*

The purpose of this section is to compare our approaches *Singleshot* and *Singleshotinexact* with *Tensorsketchonline*, which is one of the most recent divide-and-conquer type method. For *Tensorsketchonline* method, the 'online character' is artificially generated by splitting the tensor at hand into subtensors with respect to the first and the second modes.

The results of this comparison are reported in the figures 1 and 2 for the Enron and the Moovielens data sets. The methods compared achieves similar errors on the test set (different from the training set) with a slight advantage for *Singleshot* and *Singleshot-inexact* with positivity constraints on the latent factors for the Enron data set (see figure 1) and the unconstrained methods *Singleshotinexactunconstrained* and *Singleshotunconstrained* for the Moovielens data set (see figure 2). However, our approach requires less running time compared to *Tensorsketchonline* certainly due to the fact that a complete alternate minimization process is more time-consuming with respect to coordinate gradient descent.

### 4.2.1 *Singleshotonline* vs *TensorSketchonline* (Toy)

The problem at hand is to reconstruct a 3-order noisy tensor $\boldsymbol{\mathcal{Y}} = \boldsymbol{\mathcal{X}} + \sigma \times \mathcal{N}$ ($\sigma = 10^{-1}$, the entries of the real tensor $\boldsymbol{\mathcal{X}}$ are drawn from a normal distribution with zero mean and standard deviation 1 and the entries of the noise tensor $\mathcal{N}$ are drawn from a normal distribution with zero mean and standard deviation $\frac{1}{2}$) of one billion entries splitted in the following way $10000 \times 1000 \times 100$. For the latent factors inference task, we first consider an initial chunk $\boldsymbol{\mathcal{Y}}_0$ of size $1000 \times 200 \times 20$, followed by chunks of size $3000 \times 200 \times 20$, $4000 \times 200 \times 20$, $4000 \times 400 \times 40$, $3000 \times 400 \times 60$, $7000 \times 200 \times 60$, $7000 \times 600 \times 40$, $3000 \times 600 \times 100$, $10000 \times 400 \times 100$. As we consider the case where the newly streamed data is coherent with respect to the past dimensions, the subtensors have different sizes. It corresponds to updating the representation with respect to a chosen mode in the following order: mode 1 (3000 subtensors), mode 2 (200 subtensors), mode 3 (40 subtensors), mode 1 (3000 subtensors), mode 2 (200 subtensors), mode 3 (40 subtensors), mode 1 (3000 subtensors), mode 2 (400 subtensors).

*Singleshotonline* is compared with *Tensorsketchonline*. The initial points (drawn from a Gaussian distribution) as well as the stopping criterion are identically chosen. The stopping criterion is : either the fitting of the current streamed data is inferior to a fixed threshold or a maximum number of iterations is reached. The evaluation criteria are the running time and the approximation error *AE* defined by:

$$AE = \|\boldsymbol{\mathcal{X}} - \boldsymbol{\mathcal{G}}_{out} \times_1 \mathbf{A}_{out}^{(1)} \times_2 \mathbf{A}_{out}^{(2)} \times_3 \mathbf{A}_{out}^{(3)}\|_F \tag{26}$$

with $\boldsymbol{\mathcal{G}}_{out}, \mathbf{A}_{out}^{(1)}, \mathbf{A}_{out}^{(2)}, \mathbf{A}_{out}^{(3)}$ being the factors inferred from the decomposition of the noisy tensor $\boldsymbol{\mathcal{Y}}$.

Figure 1: Top: error on the test set for the Enron data set, Bottom: CPU running time in seconds for the Enron data set. The rank of the core tensor is fixed to $(5, 5, 5)$

Figure 2: Top: error on the test set for the Moovie data set, Bottom: CPU running time in seconds for the Moovie data set. The rank of the core tensor is fixed to $(5, 5, 5)$

Figure 3 presents the running time and the fitting error *AE* over three different noises. It shows that *Singleshot* slightly performs better than *Tensorsketch* while being two times faster.

### 4.2.2 *Non-negative Singleshotonline* vs *TensorSketchonline* on Movielens (this comparison for Enron has been already presented in the section 5 of the main paper)

The task at hand is a rating prediction problem. The *Movielens* dataset encompass users rating on movie along the time. We consider a tensor $\boldsymbol{\mathcal{X}} \in \mathbb{R}^{15000 \times 2000 \times 60}$ (60 users, 2000 movies). We split the data along the time-mode (70% training, 30% test) and estimate the rating on the test part given

Figure 3: Online toy problem: Approximation error in $log_{10}$ scale (left) and CPU running time (right) for various core tensor ranks. The rank of the core tensor $\mathcal{G}$ is fixed to $(R, R, R)$

Figure 4: **Movie-lens** rating prediction : Mean Approximation Error (left), Root Mean Square Error (center), Running time (right). The rank of the core tensor $\mathcal{G}$ is fixed to $(R, R, R)$

the latent factors infered for the users and the movies modes. The *MAP* (Mean Average Precisions) as well as the running time (for both of them, the value displayed is a mean over 5 different splits) are given by the figure 4. Our online approach *Singleshotonline* outperforms *Tensorsketchonline* both in terms of *MAP* while requiring less running time, which is expected as the rating prediction benefits from the non-negative constraints of our approach.

## 4.3 Assumptions check

In this part, we check in the experiments some of the assumptions made.

### 4.3.1 Evolution of the gradient with respect to the variable $\mathbf{A}^{(N)}$ (Assumption 3.2. for *Singleshotinexact*)

First, we check the evolution of the norm of (inexact) derivatives of the latent factors. The figure 5, representing the norms of $\partial_{\mathbf{A}^{(N)}} f(\mathcal{G}_{k+1}, \left\{ \mathbf{A}_{k+1}^{(m)} \right\}_1^{N-1}, \mathbf{A}_k^{(N)})$ and $\sum_{j \in \mathcal{SET}_k} \partial_{\mathbf{A}^{(N)}} \ell_j$ shows that **Assumption 3.2** for *Singleshotinexact* is valid in practice and $\eta_k^N$ well defined.

### 4.3.2 Assumption 3.2 Choice of $\mathcal{SET}_k$

A natural way to ensure the non-nullity of the inexact gradient would be to perform an exhaustive search to determine the best $\mathcal{SET}_k$ of cardinality $B_k$. This is impractical even for small values of $I_n$ because in the worst case, it would require $\frac{I_n!}{B_k!(I_n - B_k)!}$ inexact gradient computations. Thus, to get

Figure 5: Enron data set $\mathcal{X} \in \mathbb{R}^{1200 \times 1200 \times 200}$: left (gradient norm for *Singleshot* on logarithmic basis), right (inexact gradient norm for *Singleshotinexact* on logarithmic basis)

a convenient $\mathcal{SET}_k$, we prove numerically that it is sufficient to perform a random selection of the subtensors involved in the computation of the inexact gradient as proved in the section 4.3.1

### 4.3.3 Influence of the number of subtensors

Figure 6: Decomposition accuracy with the core rank fixed to $(10, 10, 10)$ with respect to the number of subtensors for *Singleshotinexact* on the *Enron* data set: left (unconstrained), right (non-negativity constraint)

The figure 6 investigates the evolution of the approximation error with respect to the number of subtensors used for *Singleshotinexact*. We notice that the more the number of slices is important, the less the approximation error. This is coherent with the intuition since a small number of subtensors induces an important error on the descent direction, and thus, lead the algorithm far from good minima.

## 5 Why the approach works for subtensors with respect to every mode

Let's consider a simple case of a three-order tensor $\mathcal{X} \in \mathbb{R}^{I_1 \times I_2 \times I_3}$. Let's assume

$$\{1, ..I_1\} = \theta_1^{(1)} \cup \theta_2^{(1)}, \qquad \{1, ..., I_2\} = \theta_1^{(2)} \cup \theta_2^{(2)}, \qquad \{1, ..., I_3\} = \theta_1^{(3)} \cup \theta_2^{(3)},$$

with:
$\theta_1^{(n)} = \left\{ 1, \cdots, \text{Int} \left( \frac{I_n}{2} \right) \right\}, \theta_2^{(n)} = \left\{ \text{Int} \left( \frac{I_n}{2} \right) + 1, \cdots, I_n \right\}$, Int(x): the greatest integer that is less or equal to x

As the discrepancy can be rewritten along the sets

$$f \left( \mathcal{G}, \left\{ \mathbf{A}^{(m)} \right\}_1^N \right) = \| \mathcal{X} - \mathcal{G} \times_{m \in I_3} \mathbf{A}^{(m)} \|_F^2$$

$$= \sum_{m_1=1}^2 \sum_{m_2=1}^2 \sum_{m_3=1}^2 \| \mathcal{X}_{\theta_{m_1}^{(1)}, \theta_{m_2}^{(2)}, \theta_{m_3}^{(3)}} - \mathcal{G} \times_1 \mathbf{A}^{(1)}_{\theta_{m_1}^{(1)},:} \times_2 \mathbf{A}^{(2)}_{\theta_{m_2}^{(2)},:} \times_3 \mathbf{A}^{(3)}_{\theta_{m_3}^{(3)},:} \|_F^2$$

The derivative of $f$ with respect to $\mathbf{A}^{(1)}$ is equal to $[\partial_{\mathbf{A}^{(1)}_{\theta_1^{(1)},:}} f, \partial_{\mathbf{A}^{(1)}_{\theta_2^{(1)},:}} f]$ ([,]: row-wise stacking

operator).
The derivative of $f$ with respect to $\mathbf{A}^{(1)}_{\theta_1^{(1)},:}$ requires the processing of the subtensors

$\left\{ \boldsymbol{\mathcal{X}}_{\theta_1^{(1)},\theta_{m_2}^{(2)},\theta_{m_3}^{(3)}} \right\}_{1 \le m_2,m_3 \le 2}$. The same reasoning obviously holds for other derivatives.

Our approach applied to this splitting scheme simply requires the sequential processing of subtensors $\boldsymbol{\mathcal{X}}_s \in \mathbb{R}^{\frac{I_1}{2} \times \frac{I_2}{2} \times \frac{I_3}{2}}$.

More generally, for $\boldsymbol{\mathcal{X}} \in \mathbb{R}^{I_1 \times \cdots \times I_N}$ with $I_n = \dot{\bigcup}_{k=1}^{p} \left\{ (k-1) \times Int(\frac{I_n}{p}) + 1, ....k \times Int(\frac{I_n}{p}) \right\}$ (disjoint union), our approach simply requires the processing of subtensors of size $Int(\frac{I_1}{p}) \times .... \times Int(\frac{I_N}{p})$.

**This means that for our approach, we can use the subtensors as small as we want since we can choose p as large as we want**.

## 6 Space and time complexity analysis

The complexity (in time) of *Singleshot* and *Singleshot-inexact* are given by the table 1. Compared to the two standard decomposition approaches Tucker-ALS (also named *HOOI*) and *HOSVD*, which use the whole tensor at once and have a complexity $\mathcal{O}(I^{N+1})$ [3], *Singleshot* requires more computations, but is more flexible than *HOOI* and *HOSVD* in the sense that *Singleshot* performs the inference task by sequential processing of small chunks of data (instead of using the whole data set at once) and can be easily extended to incorporate some popular constraints in tensor decomposition such as non-negativity.

In terms of complexity, the approaches *Singleshotinexact* and *Singleshot* differ only in computation time (see table 2) since the individual terms in the derivatives are identical (resulting in the same space complexity: see table 1), but unlike *Singleshot*, *Singleshotinexact* simply drops some of the terms instead of using all of them (resulting in a smaller complexity in time compared to *Singleshot*).

| *Singleshot* | | |
|---|---|---|
| Steps / Constraints | Update of $\mathbf{A}^{(p)}, 1 \le p \le N$ | Core update |
| Unconstrained | $(N-1)(\prod_{k=1}^{N} I_k)(\prod_{k \ne n} J_k) + I_n \prod_{k=1}^{N} J_k + I_p J_p + J_p \prod_{k \ne n} I_k^2 + I_p J_p^2 + 2I_p J_p$ | $N \prod_{k=1}^{N} I_k J_k + 2 \prod_{k=1}^{N} J_k + N \prod_{k=1}^{N} J_k^2 + \sum_{n=1}^{N} I_n J_n^2$ |
| Nonnegativity | $(N-1)(\prod_{k=1}^{N} I_k)(\prod_{k \ne n} J_k) + I_n \prod_{k=1}^{N} J_k + I_p J_p + J_p \prod_{k \ne n} I_k^2 + I_p J_p^2 + 3I_p J_p$ | $N \prod_{k=1}^{N} I_k J_k + 3 \prod_{k=1}^{N} J_k + N \prod_{k=1}^{N} J_k^2 + \sum_{n=1}^{N} I_n J_n^2$ |
| *Singleshotinexact* | | |
| Unconstrained | $(N-1)(b \prod_{k \ne n} I_k)(\prod_{k \ne n} J_k) + b \prod_{k=1}^{N} J_k + b J_p + J_p \prod_{k \ne n} I_k^2 + b J_p^2 + 2b J_p$ | $Nb \prod_{k \ne n} I_k \prod_{k=1}^{N} J_k + 2 \prod_{k=1}^{N} J_k + N \prod_{k=1}^{N} J_k^2 + \sum_{m \ne n} I_m J_m^2 + b J_p^2$ |
| Nonnegativity | $(N-1)(b \prod_{k \ne n} I_k)(\prod_{k \ne n} J_k) + b \prod_{k=1}^{N} J_k + b J_n + J_p \prod_{k \ne n} I_k^2 + b J_p^2 + 3b J_p$ | $Nb \prod_{k \ne n} I_k \prod_{k=1}^{N} J_k + 3 \prod_{k=1}^{N} J_k + N \prod_{k=1}^{N} J_k^2 + \sum_{m \ne n} I_m J_m^2 + b J_p^2$ |
| *Singleshotonline* | | |
| Unconstrained | $(N-1)(b \prod_{k \ne n} I_{k,t})(\prod_{k \ne n} J_k) + b \prod_{k=1}^{N} J_k + b J_p + J_p \prod_{k \ne n} I_{k,t}^2 + b J_p^2 + 2b J_p$ | $Nb \prod_{k \ne n} I_{k,t} \prod_{k=1}^{N} J_k + 2 \prod_{k=1}^{N} J_k + N \prod_{k=1}^{N} J_k^2 + \sum_{m \ne n} I_{m,t} J_m^2 + b J_p^2$ |
| Nonnegativity | $(N-1)(b \prod_{k \ne n} I_{k,t})(\prod_{k \ne n} J_k) + b \prod_{k=1}^{N} J_k + b J_n + J_p \prod_{k \ne n} I_{k,t}^2 + b J_p^2 + 3b J_p$ | $Nb \prod_{k \ne n} I_{k,t} \prod_{k=1}^{N} J_k + 3 \prod_{k=1}^{N} J_k + N \prod_{k=1}^{N} J_k^2 + \sum_{m \ne n} I_m J_m^2 + b J_p^2$ |

Table 1: Complexity in time per update (one iteration of gradient descent). For *Singleshot* and *Singleshotinexact*, we consider one-mode subtensors drawn from a tensor $\boldsymbol{\mathcal{X}} \in \mathbb{R}^{I_1 \times \cdots \times I_N}$. For *Singleshotonline*, we consider $b$ subtensors $\boldsymbol{\mathcal{X}} \in \mathbb{R}^{I_{1,t} \times .. I_{n-1,t} \times b \times I_{n+1,t} .. \times I_{N,t}}$ acquired with respect to a mode $n$ at the time step $t$. For all of the methods, we consider to have a core tensor $\boldsymbol{\mathcal{G}} \in \mathbb{R}^{J_1 \times \cdots \times J_N}$ and for the non-negativity, we consider the projected gradient descent [14]. with $I_{j,t}$ representing the dimension at the time step $j$

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

| Singleshot/Singleshotinexact | | |
|---|---|---|
| Steps / Constraints | Update of $\mathbf{A}^{(p)}$, $1 \leq p \leq N$ | Core update |
| Constrained or unconstrained | $\prod_{m\in\mathbf{I}_{N\neq n}} I_m + \sum_{m\in\mathbf{I}_{N\neq m}} I_m J_m + J_p + I_p J_p$ | $\prod_{m\in\mathbf{I}_{N\neq n}} I_m + \sum_{m\in\mathbf{I}_{N\neq m}} I_m J_m + J_p + \prod_{m\in\mathbf{I}_N} J_m$ |
| Singleshotonline | | |
| Steps / Constraints | Update of $\mathbf{A}^{(p)}$, $1 \leq p \leq N$ | Core update |
| Constrained or unconstrained | $\prod_{m\in\mathbf{I}_{N\neq n}} I_{m,t} + \sum_{m\in\mathbf{I}_{N\neq m}} I_{m,t} J_m + J_p + I_{p,t} J_p$ | $\prod_{m\in\mathbf{I}_{N\neq n}} I_{m,t} + \sum_{m\in\mathbf{I}_{N\neq m}} I_{m,t} J_{m,t} + J_p + \prod_{m\in\mathbf{I}_N} J_m$ |

Table 2: Complexity in space. For *Singleshot* and *Singleshotinexact*, we consider one-mode subtensors drawn from a tensor $\boldsymbol{\mathcal{X}} \in \mathbb{R}^{I_1 \times \ldots \times I_N}$. For *Singleshotonline*, we consider $b$ subtensors $\boldsymbol{\mathcal{X}} \in \mathbb{R}^{I_{1,t} \times ..I_{n-1,t} \times b \times I_{n+1,t} .. \times I_{N,t}}$ acquired with respect to a mode $n$ at the time step $t$. For all of the methods, we consider to have a core tensor $\boldsymbol{\mathcal{G}} \in \mathbb{R}^{J_1 \times \ldots \times J_N}$ and for the non-negativity, we consider the projected gradient descent [14]. with $I_{j,t}$ representing the dimension at the time step $j$

[3] Rémy Boyer and Roland Badeau. Adaptive multilinear svd for structured tensors. *2006 IEEE International Conference on Acoustics Speech and Signal Processing Proceedings*, 2006.

[4] F. Maxwell Harper and Joseph A. Konstan. The movielens datasets: History and context. *TiiS*, 5:19:1–19:19, 2015.

[5] Alexandros Karatzoglou, Xavier Amatriain, Linas Baltrunas, and Nuria Oliver. Multiverse recommendation: n-dimensional tensor factorization for context-aware collaborative filtering. In *RecSys*, 2010.

[6] Tamara G. Kolda and Brett W. Bader. Tensor decompositions and applications. *SIAM Review*, 51(3):455–500, September 2009.

[7] Jean Kossaifi, Yannis Panagakis, and Maja Pantic. Tensorly: Tensor learning in python. *arXiv*, 2018.

[8] Amy N. Langville and William J. Stewart. The kronecker product and stochastic automata networks. *J. Comput. Appl. Math.*, 167(2):429–447, 2004.

[9] Xingguo Li, Tuo Zhao, Raman Arora, Han Liu, and Mingyi Hong. On faster convergence of cyclic block coordinate descent-type methods for strongly convex minimization. *J. Mach. Learn. Res.*, 18(1):6741–6764, 2017.

[10] Xiangru Lian, Yijun Huang, Yuncheng Li, and Ji Liu. Asynchronous parallel stochastic gradient for nonconvex optimization. In *Proceedings of the 28th International Conference on Neural Information Processing Systems - Volume 2*, NIPS'15, pages 2737–2745, 2015.

[11] Jean-Paul Penot. Elements of differential calculus (chapter 2 of the book: Calculus without derivatives). *Springer*, 2013.

[12] Anthony Man-Cho So and Zirui Zhou. Non-asymptotic convergence analysis of inexact gradient methods for machine learning without strong convexity. *Optimization Methods and Software*, 32(4):963–992, 2017.

[13] Yangyang Xu and Wotao Yin. A block coordinate descent method for regularized multiconvex optimization with applications to nonnegative tensor factorization and completion. *SIAM J. Imaging Sciences*, 6:1758–1789, 2013.

[14] Rafal Zdunek and al. Fast nonnegative matrix factorization algorithms using projected gradient approaches for large-scale problems. *Intell. Neuroscience*, 2008:3:1–3:13, 2008.