[Reviews · NeurIPS 2019]

Reviewer 1



Relevant references to *linear* numerical optimization papers/textbooks that employ gradient descent to compute orthonormal matrices for the matrix SVD that this method parallels and generalizes are not given, or not clearly shown. Its main claimed advantage over other scalable tensor decompositions is its ability to extend its algorithm to handle common constraints, such as non-negativity, but it has not been discussed relative to which other work. This is a complete piece of work, but it is unclear why is this paper submitted to NIPS and not to a numerical optimization conference/journal.

Reviewer 2



Summary: The paper proposes two algorithms for scalable Tucker decomposition that are based on the coordinate gradient descent and sequential processing of data chunks. The convergence analysis is also provided. The experimental results show the the proposed method can decompose bigger tensors than competitors without compromising efficiency. The strengths are 1 The proposed scalable algorithms for Tucker decomposition could be useful in large-scale real problems. 2 A complete theoretical results of convergence analysis are provided. 3 the paper is clearly written. The main drawback of the paper is that the experimental results are too simple and seem not quite convincing. More experimental setting should be tested and more deep analysis of the results should be provided. One question is, in the experiment, will the method still work if the dimension M exceed a certain number like 10,000?

Reviewer 3



I have read the authors authors rebuttal; I still believe that the experiments are not convincing enough and that a paper claiming to beat the state-of-the-art in scalable tensor decompositions should contain more thorough and clearer experiment setup (larger-scale experiments, fewer ambiguities about experiment setup decisions). --- The authors propose a scalable Tucker tensor decomposition and a streaming extension to this method. The authors demonstrate how the memory blow-up occurring during the Tucker decomposition can be avoided, by formulating the derivations in a way that sub-tensor partitions of the original input tensor can be processed in a sequential manner. The authors provide convergence guarantees of their algorithm to a point where the gradient is equal to zero. Although the approach is sensible and it is backed by theoretical analysis, I find that the experiments are not convincing enough for a work in which improved scalability is regarded as the main advancement. I first list my comments regarding the experiments and provide with additional comments below: - The authors choose to “arbitrarily set the size of the Movielens and Enron tensors to M × M × 200 and M × M × 610”. It is unclear why they have chosen to sample the original tensors (e.g., take the 610 most frequent words). Is there an issue with the algorithm or the method proposed if all 3 modes of a 3-order tensor grow simultaneously to relatively large sizes? - Both of the real datasets used seem to be sparse — if that is the case, have the authors tried to exploit sparsity in terms of the input data and intermediate computations? If all computations target dense data and the input is sparse, the experiment results may not be representative of the full potential of the approaches under comparison. There are several approaches targeting sparse input datasets which the authors have not compared against, e.g., - Oh, Sejoon, et al. "Scalable tucker factorization for sparse tensors-algorithms and discoveries." 2018 IEEE 34th International Conference on Data Engineering (ICDE). IEEE, 2018. - Smith, Shaden, and George Karypis. "Accelerating the tucker decomposition with compressed sparse tensors." European Conference on Parallel Processing. Springer, Cham, 2017. I was left wondering how would such approaches perform for the input data considered. - Since scalability and memory blow-up issues are examined (as the central topic of the paper), I would expect the experiments to be conducted on a node with much larger memory capacity (the authors report that they used a single node with 32GB of memory). Overall experiment setup needs much more thorough description. Motivation / writing issues - Poor motivation of the problem in the Introduction; I would suggest improving the description on why scalable static Tucker decomposition is an important problem (ideally, connecting the discussion with the experiments conducted and their corresponding application domains). - Also, the authors mention that “From a streaming tensor decomposition point of view, our Singleshot extension is competitive with its competitor.” as part of the main contributions, which is confusing. If the streaming extension of the method proposed is just competitive and not superior as compared to baselines, then it should not be stated as a contribution. - Section 3.1 needs to be more specific to the challenges raised by the Tucker memory blowup problem; in the manuscript’s current version, it is not clear to the reader where the memory blowup occurs and why this is a challenge. - How is the user expected to choose the fixed mode, which is supposed to be predefined (as reported in Section 3.1)? There should be some guidance provided for such choices. Related work - The authors need to be more specific with respect to limitations of prior work; the statement “One major limitation related to these algorithms is their lack of genericness (i.e. they cannot be extended to incorporate some constraints such as non-negativity)” is vague and needs to be more precise w.r.t. each one of the prior works referred.

[Author Response · NeurIPS 2019]

The main objective of this paper is to propose a novel scalable Tucker-based tensor decomposition algorithm and more importantly to provide a convergence rate for such an algorithm. As far as we know, this is the first result of this kind especially for the exact and inexact gradient computations.

The experimental results presented in the main paper and in the supplementary material supports our claims of scalability and efficiency

**Reviewer 1.**   Our main contribution is the scalable algorithm and its theoretical analysis. Adaptivity of the algorithm to constraints is a by-product of the numerical scheme we propose not our main contribution.

Experiments using non-negative constraints are exposed in Figure 2 for both Enron and Movielens datasets. Our comparison show that our approaches can handle larger tensors than competitors. The online capability of our approach has also been illustrated on an online setting in Figure 3.

We have already more than twenty references related to tensor decompositions including papers using SVD and HOSVD but we will be happy to add some other relevant references we have missed and that the reviewer will explicitly point us to.

**Reviewer 2.**   Thanks for acknowledging the strengths of the paper. Again, we want ot stress that as far as we known, we are the first to provide convergence rate on Tucker decomposition algorithm using exact and inexact gradient descent.

Unfortunately, because we have focused our paper on the theoretical results, most experiments have been deported to the supplementary materials. Let us point to some important results

- For $M = 10000$, our approach would still be applicable. As explained in remark 1 and section 3.4, instead of computing gradients from slices of the tensor, we can decompose such a slice in subtensors and then apply our scheme on those subtensors. The Section 5 of the supplementary material provides the mathematical details of this point.
- For large $M$, we can also consider an online setting as exposed in the last paragraph of the experimental section. Results for this setting is given in Figure 3 and several online (with positive constraints) results are also available in the supplementary material Section 4.3.
- Other experimental analyses provided in the appendix deal with sanity-check of our hypotheses for the inexact gradient approach, some online decomposition performances with respect to rank of the core tensor.

**Reviewer 3**   As for reviewer 2, thanks for acknowledging that our algorihm and the provided theoretical results on its convergence rate is a strong contribution.

Regarding the experiments, we have decided to sample the original tensors of Movielens and Enron in order to be able to analyze how different competitors behave with increasing size of subtensors. The choice of the third dimension has been made especially for the competitors not to blow-up memory. As we have stated above for reviewer 2, in our case, our algorithm can handle any size of subtensors (as illustrated for instance in Figure 3 for the online setting).

Our experiments have been run on Enron and Movielens and consider as a competitor *TensorSketch* algorithm ([4] Becker et al., Nips 2018) which is tailored for sparse tensors (See their experiments in section 4).

Note also that several experimental analysis have been deported to the supplementary material due to the lack of space and the focus on the theory. Results presented there include non-negative decomposition and online decompositions ...

We have limited ourself to a single node with 32Gb of memory due to lack of better material. However, we believe that our theory and the experiments we carried out is sufficient for making the point that our approach can handle larger tensors owing to our subtensor approach.

Tensor decomposition usually leads to memory blow up during the gradient computation as they involves kronecker matrix products that have high computational spatial complexity.

The fixed mode $n$ as described in Section 3.1 can be chosen arbitrarily. The convergence of the algorithm does not depend on this parameter. However, a relevant heuristic choice is to choose $n$ such that $I_n$ is the largest dimension since this will result in the smallest subtensors to handle.

For the final version, we will improve writings and motivate better the problem we address and improve the illustration of our main contribution.

[Meta-Review · NeurIPS 2019]

The paper proposes efficient methods for computing the Tucker decomposition of higher-order tensors. The problem is a hard, basic problem in numerical linear algebra with reasonably wide applicability. Tensor decompositions have played an important role in a variety of machine learning applications, see for example: Anandkumar et al “Tensor Decompositions for Learning Latent Variable Models” JMLR 2014; Novikov et al “Tensorizing Neural Networks” NeurIPS 2015, which used tensor decompositions to massively compress the dense layers of VGG; Moitra and Wein “Spectral Methods from Tensor Networks”; and Becker and Osman “Low rank Tucker decompositions of large tensors using tensorsketch” NeurIPS 2018. Singleshot is a coordinate descent based algorithm which applies gradient updates to variables in the Tucker decomposition, which it cycles over. The paper carefully considers the memory usage of Singleshot (and its variants) since tensor computations are often extremely memory intensive. The paper proposes two related algorithms, Singleshotinexact and Singleshotonline, which reduce the computational complexity at the expense of some accuracy and compute tensor decompositions in a single pass respectively. The paper also introduces “positive” variants of the Singleshot and Singleshotinexact, for computing non-negative tensor decompositions. Although this sounds like a long laundry list of algorithms, they are all simple and natural variants on a basic theme. The analysis and algorithms hang together well. The supplementary material of 27 pages includes: an extensive analysis of Singleshot and its variants; additional experiments including results that show Singleshot’s runtime scales better with dimension than Tensorsketchonline; and a detailed discussion of the space and time complexity of the three variants of Singleshot. Section 1 of the Supp has stand-alone value as a useful reference on tensor computations. The paper tackles an important question, proposes a simple and extremely flexible (family of) algorithms, presents a detailed analysis of them, and finally presents compelling experimental results. While there could always be more experiments, more motivation, and more everything, IMO this paper is well above the bar for acceptance at NeurIPS. One of the reviewers suggested the paper is better suited to a numerical methods journal. I agree -- but it is up to the authors to choose their audience. And presumably this work will be submitted to a suitable journal soon as well.